# Managing Perceived Loneliness and Social-Isolation Levels for Older Adults: A Survey with Focus on Wearables-Based Solutions

**DOI:** 10.3390/s22031108

**Published:** 2022-02-01

**Authors:** Aditi Site, Elena Simona Lohan, Outi Jolanki, Outi Valkama, Rosana Rubio Hernandez, Rita Latikka, Daria Alekseeva, Saigopal Vasudevan, Samuel Afolaranmi, Aleksandr Ometov, Atte Oksanen, Jose Martinez Lastra, Jari Nurmi, Fernando Nieto Fernandez

**Affiliations:** 1Faculty of Information Technology and Communication Sciences, Tampere University, 33720 Tampere, Finland; daria.alekseeva@tuni.fi (D.A.); aleksandr.ometov@tuni.fi (A.O.); jari.nurmi@tuni.fi (J.N.); 2Faculty of Social Sciences, Tampere University, 33720 Tampere, Finland; outi.jolanki@tuni.fi (O.J.); outi.tamminen@tuni.fi (O.V.); rita.latikka@tuni.fi (R.L.); atte.oksanen@tuni.fi (A.O.); 3Faculty of Built Environment, Tampere University, 33720 Tampere, Finland; rosana.rubiohernandez@tuni.fi (R.R.H.); fernando.nieto@tuni.fi (F.N.F.); 4Faculty of Engineering and Natural Sciences, Tampere University, 33720 Tampere, Finland; saigopal.vasudevan@tuni.fi (S.V.); samuel.afolaranmi@tuni.fi (S.A.); jose.martinezlastra@tuni.fi (J.M.L.)

**Keywords:** architecture/built environments, gerontology, loneliness, Machine Learning (ML), mobility patterns, multidisciplinarity, sensors, social isolation, social psychology, Information and Communications Technology (ICT), wearables, wireless positioning

## Abstract

As an inevitable process, the number of older adults is increasing in many countries worldwide. Two of the main problems that society is being confronted with more and more, in this respect, are the inter-related aspects of feelings of loneliness and social isolation among older adults. In particular, the ongoing COVID-19 crisis and its associated restrictions have exacerbated the loneliness and social-isolation problems. This paper is first and foremost a comprehensive survey of loneliness monitoring and management solutions, from the multidisciplinary perspective of technology, gerontology, socio-psychology, and urban built environment. In addition, our paper also investigates machine learning-based technological solutions with wearable-sensor data, suitable to measure, monitor, manage, and/or diminish the levels of loneliness and social isolation, when one also considers the constraints and characteristics coming from social science, gerontology, and architecture/urban built environments points of view. Compared to the existing state of the art, our work is unique from the cross-disciplinary point of view, because our authors’ team combines the expertise from four distinct domains, i.e., gerontology, social psychology, architecture, and wireless technology in addressing the two inter-related problems of loneliness and social isolation in older adults. This work combines a cross-disciplinary survey of the literature in the four aforementioned domains with a proposed wearable-based technological solution, introduced first as a generic framework and, then, exemplified through a simple proof of concept with dummy data. As the main findings, we provide a comprehensive view on challenges and solutions in utilizing various technologies, particularly those carried by users, also known as wearables, to measure, manage, and/or diminish the social isolation and the perceived loneliness among older adults. In addition, we also summarize the identified solutions which can be used for measuring and monitoring various loneliness- and social isolation-related metrics, and we present and validate, through a simple proof-of-concept mechanism, an approach based on machine learning for predicting and estimating loneliness levels. Open research issues in this field are also discussed.

## 1. Introduction and Problem Statement

Social isolation and feelings of loneliness (i.e., what we call “perceived loneliness”) have had an increasing trend over the last decade, especially in countries with fast aging populations and, notably, as a consequence of the past two years of dealing with the Coronavirus Disease 2019 (COVID-19) threat [1]. More than 25% of the world population is expected to be above 60 years of age by 2050, according to the World Health Organization (WHO) [2]. In addition, according to [3], already, in 2018, more than 30% of older adults (ages 65 or higher) in Europe (EU-27 region) were living alone, in single-inhabitant dwellings, and this percentage has been increasing over the past four years.

Today, the increased perception of social isolation or loneliness is typically associated with an increased likelihood of mortality [4], depression [5,6], other negative psychological outcomes [7], and poorer cardiovascular health [8]. Thus, these factors are putting the sustainability and equity of the public health and social-care systems at risk. Therefore, innovative digitalized solutions are sought to monitor and to help in taking proactive steps to alleviate loneliness and social isolation. Such future digitalized networks will likely rely heavily on wearable devices carried by older adults due to their ease of use and portability. Such wearables will ensure wireless inter-connectivity either in a Device-to-Device (D2D) manner or supported by an Edge or Cloud computing infrastructure [9], as we will discuss later in our paper. Introducing telepresences and social robots into the wireless architectural chain will increase the system complexity, especially under dynamic wireless channels and continuously changing constraints of the scenarios.

Some of the **main challenges** of designing digitalized solutions and wearable architectures toward the monitoring and management of the perceived loneliness level in older adults are (i) finding relevant metrics and (wearable) devices to measure the loneliness and social-isolation levels; (ii) building adequate data-driven/machine-learning approaches to be able to predict dangerous levels (e.g., extreme isolation or high loneliness) based on sensor-extracted data, such as mobility features and other behavioral or socio-physiological patterns; (iii) creating reliable, efficient, and optimal wireless network architectures to support fast information exchange, data analysis, monitoring, and proactive management solutions; and (iv) supporting seamless and intelligent location-based services to the older adults for prompt first-aid support, enhanced social networking, and an active lifestyle to alleviate feelings of loneliness. However, these technological challenges must take into account the constraints and characteristics coming from other domains, such as architecture (e.g., building spaces and neighborhoods), social psychology (e.g., subjective metrics, intervention studies), and gerontology (e.g., age-related decrease in physical and/or cognitive abilities). To the best of our knowledge, multidisciplinary studies and surveys of loneliness and isolation, comprising the four aspects of technology, social psychology, gerontology, and architecture, are still missing from the current literature.

Our paper offers a **comprehensive survey** of the challenges mentioned and envisaged solutions for loneliness monitoring and management, focusing, in particular, on data that can be collected from various wearable devices, as well as on wearable architectures, which may be suitable in various living spaces. The focus is on older adult profiles, such as single dwellers in buildings with and without shared spaces, dwellers in autonomous elderly homes with multiple health services, etc.

Thus, the main research questions addressed in this paper are:What is understood by loneliness and social isolation from the multidisciplinary perspectives of technology, social psychology, gerontology, and architecture/building environment?What is the interplay between the four aforementioned domains, and how can they converge toward enhanced monitoring and management solutions against loneliness and social isolation in older adults?What measures, metrics, and wearable or Internet of Things (IoT) devices are available to quantify different levels of loneliness and/or social isolation among the older population?How to apply ML techniques on data harnessed from wearables to offer solutions for loneliness monitoring, prediction, and management?

To answer these questions, our paper collects the multidisciplinary views of a team of authors formed by experts in ICT domain, architecture/built environments, gerontology, and social sciences/social psychology, to offer a comprehensive survey of challenges and solutions in utilizing various wearable sensors and technologies for monitoring and providing support in controlling the negative effects of the social isolation and perceived loneliness in older adults (e.g., above 55 years old).

The rest of the paper is as follows: Section 2 presents the main progress beyond the state of the art aimed with the current survey. Section 3 gives an overview of the literature landscape in loneliness and social isolation and addresses the four dimensions of the problem at hand, namely social-psychology, gerontology, technology, and architecture. Section 4 gives a survey of loneliness definitions and metrics, mathematical models, and inter-related measurable parameters; an illustrative example based on open-access loneliness data is also provided. Section 5 discusses various wearable-based solutions for measuring loneliness and social-isolation levels in detail. Section 6 introduces the building environment constraints and discusses the loneliness with respect to the living environment or architecture. Section 7 proposes wearable-based monitoring and management solutions relying on ML and recommendation systems and presents a proof-of-concept based on dummy data. Section 8 presents the conclusions, summarizing the open challenges, design recommendations, and the way forward.

## 2. Progress beyond the State of the Art

The progress beyond the state of the art is best illustrated in Table 1, where we compare our work with other surveys from the literature addressing loneliness monitoring and/or alleviation, as well as social isolation.

The authors in [10] focus on loneliness metrics and the relationship between the subjective feelings of loneliness and the objective social isolation. Technology aspects, ML, and sensor data types are not a part of their study.

The review in [11] studies the social-physiological aspects of loneliness and the triggered impairments by loneliness and social isolation.The use of mobile-collected sensor data to monitor behavior, loneliness, and mental health is studied in [12]. The focus on loneliness is rather low in [12], and the main focus is to demonstrate that smartphone sensing is a viable solution for remote psychiatric assessment.

In [13], ICT solutions to alleviate loneliness in elderly are discussed; the focus is on the so-called “social technology”, also known as technology that can be used for increased social well-being, and on the desirable end-user features of such technology, such as ease of use, cost, availability, etc. No particular sensor data is studied, and no wearable-based nor ML-based solutions are discussed.

Digital technology use for increased well-being of older adults is also discussed in [14], and loneliness is measured through two metrics: the sense of belonging and the self-esteem scale; no particular ICT technology, wearable, nor sensor data is discussed, and the metrics are rather qualitative, rather than quantitative, in their description.

Artificial Intelligence (AI) and ML algorithms to predict loneliness are studied in [15]. More than six ML algorithms are studied, and the best performance to quantify and predict loneliness is achieved with K-Nearest Neighbor (kNN) and Artificial Neural Networks (ANN). However, their input data is speech data; no other sensor data types are considered.

The authors in [16] study the neurobiology of loneliness and listed different sensor-based markers related to loneliness, such as various tomographies or Electroencephalography (EEG). The recent survey in [17] deals with loneliness of older adults during the COVID-19 era. No quantified metrics are used there; rather, the analysis is based on self-reported data. Technological or architectural aspects are not addressed.

Although age-friendly built environment aspects are addressed in [18], the technology aspects are not included in that study.

Previous work in [19] offers a systematic literature review on ICT solutions to deal with social isolation of older adults. Unlike the current survey, the work does not consider different sensor data or connectivity solutions, look into the loneliness and social-isolation metrics, nor address the urban architecture/built environment aspects. ML algorithms are also not addressed in [19].

## 3. Literature Landscape

Our study is approached from four dimensions, as illustrated in Figure 1, namely social psychology, gerontology, technology, and architecture/building spaces. The identified works, pertaining to each of these four dimensions, as well as to inter-related domains, are outlined in the next sub-sections. The text below each domain/cross-domain shown in Figure 1 denotes examples of research issues encompassed by each of those domains. For example, the technological design that takes into account the built environment specifications and constraints can create the so-called *‘smart city’* concept, a city with various nodes (buses, buildings, people) wirelessly inter-connected through sensors and wearables; the intersection between gerontology and social psychology domains addresses issues, such as decreased options for socializing and decreased mobility with older age, etc.

The domain-by-domain state of the art is discussed in the next sub-sections.

### 3.1. Social-Psychology Aspects

Loneliness is, fundamentally, a social-psychological phenomenon. The need for social belonging and meaningful social connection is an integral part of being human [20,21]. One of the key theories of social psychology, namely the self-determination theory, postulates that the need for relatedness is one of the basic needs, beside the need for autonomy and competence [22,23]. Every human has a social need to some extent, although the need may vary individually. In other words, some individuals, for example, may have higher social needs than others, but all individuals fundamentally have social needs.

Over the past few decades, social-psychological research has investigated the benefits of social ties and social support, which have a remarkable positive impact on psychological and physical well-being [20,24,25,26,27]. This core finding has been replicated over and over again in studies, extending over the past 100 years [20].

Especially after stressful life events, emotionally supportive interaction is important, and it is considered to buffer the harmful effects of stressful events [28,29]. The role of perceived social support—a subjective acknowledgment that the help is available when needed—has been especially found important in coping with stressful life events, even more so than the actual received social support. This underlines the role of a subjective interpretation of the use of social ties. From the perspective of social psychology, loneliness is derived from the perception of a lack of significant others. Loneliness, in short, denotes that social needs are not fulfilled, and there is the perception of a lack of a supportive environment. According to the cognitive-discrepancy model by [30], loneliness occurs when one perceives a discrepancy or mismatch between own achieved and desired or needed social relationships. More about social-psychology aspects in characterizing loneliness and social isolation is discussed in Section 4.1.

### 3.2. Gerontology Aspects—Impacts of Loneliness and Social Isolation

Loneliness is often seen as a particular problem of older adults, but not all older people are lonely [31]. The prevalence of loneliness varies between populations, welfare regimes, and cultural areas [32,33,34]. Central factors linked to loneliness in old age include high old age, living alone, marital status (e.g., widowed or divorced), health problems or chronic illnesses, and low social and/or material resources [31,32,33,34,35]. In a recent review of cross-sectional studies on predictors of loneliness among older adults conducted in [36], female gender, non-married status, older age, poor income, lower educational level, living alone, low perceived quality of social relationships, poor self-reported health, and poor functional status were found to be significantly associated with loneliness. Psychological attributes associated with loneliness included poor mental health, low self-efficacy beliefs, negative life events, and cognitive deficits [36]. In addition, in various qualitative studies, participants also mentioned environmental barriers, unsafe neighborhoods, migration patterns, inaccessible housing, inadequate socializing resources, and the recent loss of family and friends. A recent systematic review of longitudinal studies showed similar results of risk factors of loneliness linked to not having a partner or having lost a partner, having a limited social network and low level of social activity, poor self-perceived health, and suffering from depression or depressive mood [37].

At a higher age, perceived loneliness has some special features. Older individuals who suffer from mild cognitive impairments are significantly lonelier than cognitively healthy older adults [38]. Cognitive functioning declines with age, but most of the elderly population has good cognitive functioning, even the most senior ones. Still, it is relevant to notice that the oldest age groups are growing, which means that the number of people with memory problems will increase. Other risk factors identified concerning older adults’ loneliness and social isolation are sensory health problems, such as loss of hearing [39], poor health, and widowhood [40,41]. The loneliness of older people often arises from a complex combination of several overlapping factors. To summarize some key factors, loneliness among older people is associated with health status and self-perceived health, losses in a social network, particularly the loss of a partner, and material resources that create emotional loneliness but also affect indirectly through decreased options for socializing [35]. While not all older people are lonely, loneliness has been shown to have serious effects, such as the decline of well-being and functional ability and increased risk of morbidity and mortality [11,42,43]. The ongoing COVID-19 pandemic, with its meeting restrictions, has increased older people’s loneliness and decreased their well-being [44]. It can be concluded that the loneliness of older people is a serious matter, and, when developing tools or interventions to alleviate the loneliness of older adults, subjective factors, such as satisfaction with social contacts, as well as external factors, such as accessible and safe physical and social environments, need to be considered, along with objective factors, such as chronic illnesses or losses of a partner.

### 3.3. Technology Aspects

The third domain depicted in Figure 1 is related to various devices, sensors, wearables, and (wireless) connectivity-related aspects. Several mechanisms to improve social-interaction levels have been discussed in [45] and are grouped into three main categories:individual mobility-related aspects, such as barrier-free sidewalks and route patterns,social mobility-related aspects, such as shared and multi-functional spaces, andnature enjoyment, e.g., natural art design or the company of a living pet.

The first two categories related to mobility are of particular interest in this research work, as they can be associated with measurable metrics derived from wearable sensor data, e.g., sensors able to compute the user location, such as those including Global Navigation Satellite Systems (GNSS), Ultra Wide-Band (UWB), cellular (Fourth generation of cellular networks (4G), Fifth generation of cellular networks (5G), or Bluetooth Low Energy (BLE) chipsets. The third category is less straightforwardly measurable through sensors than the first two categories, though context-awareness sensors could also contribute to quantifying, to some extent, nature enjoyment.

A wearable-sensor-based analysis of the loneliness levels for college students was conducted in [46] via smartphones and Fitbit devices. The features extracted from wearables were the user’s mobility patterns (e.g., based on location estimates and step counters), sleep behaviors, smartphone usage, and data traffic (e.g., amount of data uploaded or downloaded, as well as the amount of speech data, which reflects how much time is spent in discussing with others). The users were divided into two classes (high-level of loneliness versus low-level of loneliness) based on a self-filled questionnaire and a numerical threshold set by the authors in [46], according to the answers to questionnaires. ML algorithms were employed to predict loneliness based on behavioral features or patterns, and classification accuracies higher than 80% were achieved. The study in [46] was limited to college students only, and the elderly population was not addressed. The adopted methodology of first identifying the behavioral and mobility patterns and then applying ml algorithms to predict the loneliness level is, however, likely to be scalable across all age groups, observing that the relevant patterns or features may be age-dependent.

Wearable devices can also record psycho-physiological data that can be further related to individual and social mobility. Continuous psycho-physiological data include, for example, heart rate, sleep quality, anxiety level, stress events or stress level, physical activity level, accelerometer data (also related to mobility patterns), etc. This psycho-physiological data from sensors and wearables allows monitoring and analysis of different individual aspects, including perceived loneliness and social-isolation levels. An exploratory pilot study was conducted on 22 participants in [47], which collected user’s demographic data, sleep data, beat-to-beat data (i.e., the time elapsed between two pulses in an electrocardiogram), daily activity data, ActiGraph data, heart-rate data, and questionnaire data. The study explored the descriptive statistics of various time-domain, frequency-domain, and non-linear-domain features of heart-rate parameters, sleep data, activity data, etc. Although the study was limited to statistical analysis (and no ML algorithm was applied), we postulate that the features explored there could also be used to make more generic predictions for identifying social isolation using various ML algorithms. Indeed, AI and ML algorithms can be used to analyze the sensor data, as well as categorical values, interviews, and speech text using natural language processing.

Another study was conducted in Reference [15] on 80 English-speaking older adults to quantify the sentiments and features that indicate loneliness. The transcribed speech text was examined using the Natural language processing (NLP) method. Quantitative loneliness was determined based on the University of California, Los Angeles (UCLA) score (see details in Section 4.1), and the input features included five emotions, namely: joy, fear, anger, disgust, and sadness. The features from the transcribed speech were extracted in the forms of frequency and inverse-document frequency and were given as inputs to ML algorithms. Algorithms, such as Support-Vector Machine (SVM) (e.g., linear kernel, polynomial kernel, and Radial Basis Function (RBF) kernel), artificial Neural Network (NN), K-nearest neighbor, AdaBoost, and Random Forest, were used to make predictions. More details on ML algorithms are to be given in Section 7.2. In the study of [15], by using linguistic features, artificial NN showed a better performance to predict loneliness than the other investigated ML algorithms.

Another survey was conducted among 427 respondents to determine the prevalence of loneliness and associated factors in health science students [48]. Questionnaires regarding demographics, daily activity, financial concerns, support systems, etc., along with 20 other items for UCLA loneliness scale were included. As the ML algorithm, the Logistic Regression was used to predict the binary outcome (yes or no) for the social-isolation level.

### 3.4. Architecture and Living-Spaces Aspects

In addition to the three domains mentioned in the previous three sub-sections, there is also the architectural aspect. Indeed, there exists a wide range of living alternatives, depending on the diverse and changing needs and degrees of independence of the varied societal group of the older adults, which range from the young-old to the older-old. Such living options span from the possibility of living on their own, in a life-long home and/or neighborhood, fully individually or in co-housing/co-living premises, to living in more institutionalized dwellings or so-called senior housing communities or community-dwellings. The living options can also span from living in active-adult communities or independent-living communities to living in assisted-living communities, continuous-care retirement communities, or assisted-living residences. Other options include adult family homes and sheltering or nursing homes or houses. Combining these typologies in multi-modal and flexible models is also possible, as a mixture of independent living units and external adult-care and healthcare services.

On the one hand, the institutionalized models imply, generally, that older adults must move from their neighborhood and community and start anew in a completely unknown environment for them, most of the time located away from family and friends. Thus, such an option entails the danger for the older adult of becoming socially isolated and increasingly lonelier. In contrast, it would also seem that residents in these facilities might feel less lonely and isolated than those living alone in their own homes since these facilities provide spaces and activities for socializing [49,50].

On the other hand, studies, such as the one in [51], report on how, for *“most elderly people living at home, [...] their satisfaction with their immediate surroundings increased, their life satisfaction improved, and loneliness reduced”*. Likewise, the studies in [52,53] suggest the beneficial impact that age-in-place has in avoiding social isolation and perceived loneliness in older adults. It seems that people in these conditions feel free and independent, as well as attached to a place and a community, which is developed over time, i.e., they experience the benefits of feeling a *sense of place* and a *sense of belonging*. Evidence such as this, together with the tendency of decentralization of health-and-care systems, due to economic reasons and to the shortage of caretakers, has led to the worldwide consensus of promoting the strategy of *aging-in-place*, i.e., *“the ability to live in one’s own home and community safely, independently and comfortably, regardless of age, income, or ability level”* [54].

In addition, due to the ongoing COVID-19 pandemic, the tendency toward *aging-in-place* seems to have ground for consolidation. Nevertheless, this concept still has limitations, such as, for instance, the literacy levels of older people and of their families on how to access the appropriate healthcare services, the feeling of not being cared about, and of becoming socially isolated due to loss of relatives or due to physical or mental impairments, which reduces the occasion for social interaction. All these depend on the personal and contextual circumstances of the older adult. For instance, the authors in [55] report high prevalence rates of loneliness in urban subsidized housing, which are ghettos of older people with poor health and limited economic resources. Thus, the study in [55] implies that a successful aging-in-place is very much dependent on the circumstances of the socio-economic, psychological, and physical conditions, as well as the socio-spatial characteristics of the place where the person is aging.

It seems then that none of the living arrangements, neither the independent ones nor the institutions for older adults, per se, can be considered best in terms of avoiding loneliness and social isolation of older adults. On the contrary, there are so many inter-related and diverse variables that the problem-solving approach, actually, suggests the need for customized solutions, depending on each circumstance. It also implies the need for multidisciplinary research and interventions to assess these multifaceted phenomena. There is a need for a combination of socio-spatial, gerontology, and technological solutions (e.g., Ambient-assisted living (AAL) technologies, social robots, artificial intelligence), to respond adequately to the problem. Such mixed solutions could be framed under the umbrella term of *smart age-friendly living environments*.

There are two European Innovation Partnerships (EIPs), the Smart Cities and Communities and the Active and Healthy Aging ones, that have the objective of extending smart-city principles to address older citizens’ needs. There is also an incipient idea of creating a cross-EIPs action spanning this objective to convey the concept of smart age-friendly cities to join these two ideas (smart cities and active and healthy aging) that are still perceived as separate [56]. The World Health Organization (2007, 2008) [57] defines age-friendly cities as: “*inclusive and accessible urban environments that benefit their aging populations*”, and the European Union (2021) [58] defines a smart city as: *“a place where traditional networks and services are made more efficient with the use of ICT for the benefit of its inhabitants and business”*. However, there is an alternative extended meaning of smartness, which considers the involvement of a broader range of multidisciplinary mechanisms, in addition to technology, to achieve wise management of resources and social sustainability [59], encompassing the implementation of services for the elderly population to cope with the need for more caregivers and rising costs of elderly care expenses.

In this regard, and specifically concerning older adult’s social isolation and loneliness, we could think about mixed (socio-spatial and technological) solutions to which studies suggest: if the perceived and achievable social support, together with the physical competence, are highly maintained, then the loneliness feeling decreases [51,60,61], regardless of whether the older adult lives independently or in an institution.

### 3.5. Inter-Dependencies between Domains

The relationship between technology and perceived loneliness/social isolation among older adults has been studied through various intervention studies, systematic reviews, and meta-analyses. The results of these studies are contradictory, and the connection between the use of digital technology and the alleviation of loneliness of older adults is not clear-cut.

On the one hand, the systematic reviews and meta-analysis in [62,63], on the effectiveness of digital technology solutions in reducing older adults’ loneliness and depression, showed inconsistent results, with some studies showing moderate effects on decreased loneliness and depression. The review of studies in [64], on the effect of video calls on the loneliness of older people, also found no significant effect. However, they called for studies with more rigorous methods addressing diverse target groups, a range of settings, and participants with demonstrated loneliness or social isolation.

On the other hand, according to several cross-sectional studies, ICT is useful in alleviating social isolation [65], loneliness [13], and depression [66] of older people. Frequent digital technology use has been shown to have a positive relationship to the perception of self-worth but no relationship with the sense of belonging [14]. Mobile Telepresence Robots (MTR) have also been considered as potential tools to strengthen the social relations of older adults, but some difficulties need to be overcome before implementing them at large scales, such as the sometimes-complex interfaces, requirements to allocate a non-negligible time to learn and to practice with them, and the risk of dehumanization [67]. The study in [13] looking at older adults’ use of social technology, such as using e-mail, social networking sites, online video/phone calls, online chatting/instant messaging, and/or a smartphone, found a positive association with social-technology use, higher subjective well-being, and reduced loneliness levels. The study in [68] showed that voice-assisted e-mails, instead of traditional e-mails, might enable keeping contact with other people and reduce loneliness. Innovations, such as Wii Bowling, can provide new opportunities for socializing and, as a result, minimize loneliness [69]. While the COVID-19 pandemic has increased loneliness among older adults, the rapid development of different digital e-services and applications have provided means for online socializing and, thus, decreased feelings of isolation and loneliness [17].

When researching the perceived loneliness of older adults through technology, some specific features need to be considered. For example, in Finland, a society of advanced digitalization of the whole society, still, in the age group of 75–89 year old persons, only 34% have a smartphone for their own use [70]. Among those aged 90 years and over, the percentage is likely even smaller. The COVID-19 pandemic has resulted in a ‘digital leap’ in many countries, but virtual contacts have not been possible for all older adults [71]. Thus, it can be argued that age-related digital division still exists [71,72,73]. For developing technology to meet the needs of the elderly, it is necessary to notice that many older adults have a negative and cautious attitude toward technological solutions [74]. Older people may have enhanced awareness of the shortness of life [75], hence having no interest in new technology solutions or learning how to use them. The rapid development of digitalization and the availability of different devices with different technologies may also strengthen the sense of feeling old and out-of-touch and, thus, can deepen the sense of social isolation [72]. In a similar vein, the study in [14] argued that over-dependence on technology can even reduce the feeling of belonging within older adults and, thus, can lead to an enhanced sense of social isolation.

Diverse and inconclusive results of the usefulness of technology in reducing loneliness of older adults are likely since older adults as a target group in technology studies cover people over age 55 to centenarians. Participants of the studies represent different generations, lifestyles, and cultural backgrounds, people with diverse educational, social, and economic resources, and diverse physical and cognitive health conditions. Hence, developing technology to alleviate older adults’ loneliness and social isolation requires tailored person-centered approaches that acknowledge older adults’ vast heterogeneity.

A recent systematic review [19] analyzed 23 recent and relevant studies related to new technologies applied to assess older adults’ loneliness and social isolation, stating that the field is growing rapidly. The studies compiled in the review in [19] focused on what the authors call ‘physical ICT’, i.e., robots, smart houses, and wearables. Seventeen studies contributed to the field of older adult’s loneliness research, and the rest (6 studies) investigated the social isolation or both phenomena (loneliness and social isolation) together. Detection with prediction and alleviation were the two kinds of interventions identified, with 15 papers focusing on the latter and 7 addressing the former. Among the alleviation studies, all used wireless technologies, for example, an ambient-activity stimulating system, including an activity sensor, a smart-home solution, etc. Sensor and wearable technologies were used in all the detection-with-prediction interventions to track older adults’ daily routines so that the system could infer the correlation of their daily actions with perceived loneliness or social isolation, with a basis on previous ML processes. As reported by the authors in [19], the results of the study demonstrated that loneliness and social isolation in older adults could not be eliminated only by technological means. However, technology does help in detecting, predicting, or alleviating these phenomena. Authors in [19] also reported that the current studies acknowledged that there is a need for more robust study samples and study designs and that there are still several technological challenges to be solved, as well as cost-related issues.

## 4. Loneliness Measures and Metrics

To analyze and improve the overall picture of loneliness and social isolation for older adults, there is a need to briefly discuss the related methods, measures, and underlying definitions. We start with definitions regarding loneliness and social-isolation concepts in Section 4.1, continue with identified mathematical models to quantify various metrics in Section 4.2, then provide an illustrative example based on open-access loneliness data in Section 4.3, and we finish this section with a discussion in Section 4.4 about additional measurable metrics or factors related to loneliness and social isolation.

### 4.1. Definitions and Widely-Adopted Metrics

Loneliness is commonly defined as a subjective unpleasant experience that derives from a perceived deficiency in social relationships [30]. In the typology of social and emotional loneliness, by [76], social loneliness refers to the absence or lack of a broader group of contacts, and emotional loneliness stands for absence or lack of an intimate relationship. Loneliness is a synonym for perceived social isolation but not for objective social isolation. Objective social isolation describes a quantifiable aspect of social relations, and its common markers include limited social contact with others and having few social-network ties [4]. Other related, but distinct, concepts include living alone, aloneness, and solitude [77].

A widely used, reliable, and validated measure for perceived loneliness is the UCLA loneliness scale [78,79], which has been revised and updated to versions of different lengths. A commonly used short version is the three-item UCLA loneliness scale [10], which measures perceived loneliness with the following questions: “How often do you feel” (1) “that you lack companionship”, (2) “left out”, and (3) “isolated from others?” Answers are given on a 3-point scale (1 = hardly ever, 2 = some of the time, 3 = often). The overall loneliness score is computed as the sum of the three sub-metrics, ranging on a scale from 3 to 9. A higher score indicates the presence of a higher loneliness level.

Another reliable and validated measure for perceived loneliness is the Dong Jong Gierveld Loneliness Scale (DJGLS). DJGLS can be used for measuring overall, social, and emotional loneliness [80,81]. In its short 6-item version, three negatively-formulated items gauge emotional loneliness, specifically, “I experience a general sense of emptiness”, “I miss having people around”, and “I often feel rejected”; and three positively framed questions inquiring social loneliness: “There are plenty of people I can rely on when I have problems”, “There are many people I can trust completely”, and “There are enough people I feel close to”. Answer options are typically *yes, more or less*, or *no*. For emotional loneliness, the neutral (*more or less*) and positive answers (*yes*) are summed up; thus, the scale ranges from 0 to 3. For social loneliness, the neutral (*more or less*) and negative answers (*no*) are counted, and the scale ranges from 0 to 3. The total loneliness score can be computed by summing up the emotional and social loneliness sub-metrics scores. A higher score indicates higher loneliness.

A reliable and validated instrument for measuring social isolation is the Lubben Social Network Scale (LSNS) [82,83]. The 6-item version includes three questions related to family and three to friendships. The questions related to family are: “How many relatives do you” (1) “see or hear from at least once a month”, (2) “feel at ease with that you can talk about private matters”, and (3) “feel close to such that you could call on them for help?” For friendships, the questions are: “How many of your friends do you”, (4) “see or hear from at least once a month”, (5) “feel at ease with that you can talk about private matters”, and (6) “feel close to such that you could call on them for help?” Answers are anchored on a scale from 0 to 5 (0 = none, 1 = one, 2 = two, 3 = three or four, 4 = five through eight, 5 = nine or more). A total social isolation score is the sum of these six items, ranging on a scale from 0 to 30. A higher score indicates more social engagement.

In [6], the perceived loneliness level was measured on a 5-level Likert scale from 0 (‘never’) to 4 (‘very often’). Another recent study [7] divided the loneliness metric into three sub-metrics, namely the lack of companionship, the feeling of being left out, and the feeling of being isolated from others, and measured each of these loneliness sub-metrics on a 3-level Likert scale, i.e., from level 1 (‘hardly ever’) to level 3 (‘very often’). Thus, the overall loneliness metric was computed as the sum of the three sub-metrics, on a scale of 3 to 9. These three metrics are also known as “the three-item UCLA loneliness scale” metrics.

A fascinating finding in [7], based on a survey among 4885 participants, was that the loneliness level decreases with increasing ages for the elderly (i.e., above 65 years old), and there was a significant inverse correlation between age and perceived loneliness level. Other investigated factors were the median household income (also found to be negatively correlated with the perceived loneliness level) and the socio-demographic density (found to be uncorrelated with the perceived loneliness level) [7].

### 4.2. Generic Mathematical Modeling

The loneliness index reported in [84] can be seen as a generalized version of Likert-scale indices from [6,7], and it can be summarized as
(1)$$L_{u}^{\left(t\right)}=\sum_{m=1}^{M}\frac{V_{m,u}^{\left(t\right)}W_{m,u}}{M},$$
where $L_{u}^{\left(t\right)}$ is the generalized loneliness index of user *u* at time *t*, $V_{m,u}^{\left(t\right)}$ is a self-reported *m*-th criterion (associated to loneliness) of user *u* at time *t*, given on a Likert scale 0 to *N*, with *N* even and m=1,…,M, *M* is the total number of criteria associated to loneliness, and $W_{m,u}$ is a positive or negative weight of the *m*-th criterion of the *u*-th user (assumed to be independent on time). For example, if one would adopt the 3-level Likert scale and the three loneliness criteria from [7] (i.e., the lack of companionship, the feeling of being left out, and the feeling of being isolated from others), then N=2 and M=3, and each $V_{m,u}^{\left(t\right)}$ would take values between 0 and N=2, ∀t,∀u. If, in addition, all three criteria would be assumed to influence the overall loneliness level equally, $W_{m,u}=1$, ∀m,∀u.

The relationship in Equation (Equation 1) can also be seen as a generalization of all metrics described in Section 4.1, a generalization that also takes into account the time dimension, as loneliness and social-isolation levels can vary in time.

Another measurable metric, this time related to the objective social isolation $I_{u}^{\left(T\right)}$, can be associated with the average of the distance to other persons in a time window *T* (i.e., higher isolation for larger average distances):(2)$$I_{u}^{\left(T\right)}=\int_{t=0}^{T}\sum_{j=1}^{N_{u}}d_{j,u}\left(t\right)dt,$$ where $N_{u}$ is the total number of persons encountered during a pre-defined time window *T*, and $d_{j,u}\left(t\right)$ is the physical distance between person *u* and person *j* at time *t*, as measured by a proximity sensor (e.g., GNSS-based, BLE-based, Wireless Fidelity (WiFi)-based, also known as IEEE 802.11, etc).

Thus, Equations (Equation 1) and (Equation 2) can characterize the perceived loneliness and the objective social isolation levels, respectively, in a quantitative and measurable manner, assuming one is able to collect user data about $V_{m,u}^{\left(t\right)}$ and $d_{j,u}\left(t\right)$ values. The $W_{m,u}$ weights choice is a more qualitative problem, and it may require the social-psychology perspective in choosing it adequately. While the $d_{j,u}\left(t\right)$ distances are today easily measured via positioning or proximity-detection sensors, as can be seen in recent examples in [85,86,87,88], the measurements of $V_{m,u}^{\left(t\right)}$ are less trivial and may require various underlying models and inter-dependencies of various parameters related to loneliness. Some of those parameters are discussed further in Section 4.4 and Table 2, and a survey of possible sensors to measure $V_{m,u}^{\left(t\right)}$ and $d_{j,u}\left(t\right)$ parameters is further given in Section 5.

### 4.3. Illustrative Example

In order to map Equation (Equation 1) into a concrete example, Figure 2 shows an illustration of two loneliness indices. The plots in Figure 2 were computed based on five underlying factors shown to have an impact on loneliness, and they are based on the open-access dataset provided in [89]. Unfortunately, comprehensive details about how the five $V_{m,u}^{\left(t\right)}$ parameters (see Equation (Equation 1)) were computed are missing from [89]; nevertheless, the dataset is valuable for illustrative purposes and as a basis to understand the spatial fluctuations in loneliness indices.

The five factors related to loneliness and used in [89] were: Alzheimer’s, depression, high blood pressure, anxiety, and insomnia. The complementary indices were considered to have a higher index corresponding to a higher level of loneliness. As aforementioned, the details of how those indices were computed based on the five underlying factors were not provided, but the overall numbers, collected from 6791 elderly persons in the UK and 445 elderly persons in Scotland, showed good symmetry and a rather similar distribution among the two regions, as shown in Figure 2. A 0 index corresponds to an average loneliness level; above 0 points to a high loneliness level (e.g., a slightly higher percentage of lonely persons in Scotland than in the UK, according to Figure 2), and above 0 points to a low loneliness level.

Based on the same datasets provided in [89], one can also have a more thorough spatial characterization by making use of the geographical heat maps. Such geographical heat maps display the data over a geographic area using different colors and shades. In the context of perceived loneliness among elderly people, visualization plots, such as geographical heat maps, can highlight the areas with a higher or lower loneliness index. Figure 3 shows the geographical heat map representing the areas of high or low loneliness index in England (left) and a zoomed in version of the London area (right). Based on 2019 data from [89], the value of the loneliness index in England varied between −20, indicating a very low loneliness index, and 5.5 showing a high-value loneliness index. In most areas, the loneliness index value is between −1.0 to 1.0, which indicates a moderate value for the loneliness index. In Figure 3, the areas with dark blue colors are the ones that have the lowest values of loneliness index, and the areas with dark red colors have the highest value of loneliness index.

This example emphasizes several take-away points: (i) the scalar model from Equation (Equation 1) is good enough to differentiate between classes of users (according to their loneliness levels); (ii) if sensor-collected training data is available from a high enough number of users, statistical models, ML models, or other prediction algorithms can be used to classify the users according to their loneliness levels, possibly taking into account various other constraints (e.g., geographical distributions, building environments and other context-aware constraints, etc.); and (iii) spatio-temporal information about users can be beneficial for deriving ICT-based solutions for loneliness monitoring and possible management.

### 4.4. Related Metrics

As already shown in Equation (Equation 1) and the illustrative examples of the previous sub-section, the prediction of the perceived loneliness levels requires data from various sources to be quantified in a certain amount of metrics. These metrics need underlying models in a direct or inversely proportional relationship with the loneliness index. Such metrics can be obtained from various sources, such as sensors, questionnaires, behavioral data, etc. Sensors, such as Electrocardiography (ECG) sensors, accelerometers, Photoplethysmography (PPG) sensors, smartwatches, smart rings, or proximity sensors, can record continuous data about various psycho-physiological data. The features extracted from these continuous signals can be used as loneliness predictors. The data from questionnaires or interviews can also be used as metrics for loneliness predictions. Loneliness-related metrics can be directly or inversely proportional to the perceived loneliness level and can influence the value of the loneliness index either positively or negatively. For example, the findings in [46] showed that the perceived loneliness level was inversely proportional to the activity level of each user and directly proportional with the frequency of sedentary behavior. Varied and complex underlying models of loneliness are still needed in the current literature, as also discussed later in Section 8.

Table 2 summarizes the main loneliness predictors found in the current literature and which are measurable either through questionnaire-based approaches or via sensor/wearable-based approaches. The last column in Table 2 briefly describes the corresponding metric and its potential relationship to loneliness levels. Some of the relationships have not been directly studied in the literature thus far, but they are indirectly related, e.g., Morningness-eveningness questionnaire (MEQ) metric is indirectly related to loneliness levels as explained in the last column of Table 2. In addition, the Positive and Negative Affect Schedule (PANAS) metrics, namely the Positive affect (PA) and Negative affect (NA) factors, related to neuroticism and extraversion, are indirectly related to loneliness levels.

The next section, Section 5, further addresses the metrics from this section in terms of how they can be measured and what kind of available sensors, wearables, or other ICT devices can be employed to harness such measurements.

## 5. Wearable Sensors for Measuring Loneliness and/or Social Isolation Levels

The nature by which human activities and actions are recorded and measured is of significant importance in loneliness management, and it plays a vital role in the quality of gathered information concerning environmental, behavioral, and social factors.

One important aspect of loneliness-management solutions is to predict the onset of loneliness in older adults. In particular, social isolation can be the main driver of perceived loneliness. This section gives an overview of sensors and wearables that can be used to measure various loneliness-related and social-isolation-related metrics. To select suitable solutions, a designer should consider several factors toward the choice of the ICT solution. Here, we have chosen five relevant aspects in the wearable sensor choice, and they are addressed in the following sub-sections. These five aspects are: (1) the sensor attributes, (2) the sensor obtrusiveness (also related to the ease of use), (3) the energy consumption, (4) the data extraction and manipulation, and (5) the application requirements (e.g., prediction, monitoring, management, etc.).

### 5.1. Sensor Attributes

Wearable sensors must provide multi-modal information about older adults. Such data can include, but cannot be limited to: physical state of the participant, geographical attributes of the participant (e.g., outdoors versus indoors, rural versus urban, etc.), accurate location (e.g., if indoors, this location is typically also related to the current activity), movement patterns (and body language if possible), vital signs of the participant (e.g., heart rate, skin conductivity, temperature, EEG, ECG, etc.), possibility of gathering manually added data (e.g., data added via text or voice by the user), etc. The data can be collected by multiple sensor sources that measure different physical metrics, which may or may not occur concurrently and may be continuous or discrete in time. The data frequency collected from various sensors may also vary. Various Radio Frequency (RF) sensors, which were identified based on our literature and market studies, are tabulated and compared in Table 3. The focus is on multi-functional and non-invasive wearables, able to provide more than one metric related to loneliness, as well as on low-to-moderate cost and easy-to-carry wearables. The video-based solutions are excluded from Table 3, as the user privacy levels in the presence of monitoring cameras are lower than in the presence of RF-based solutions. Table 3 is, of course, not aiming to provide a comprehensive overview of all available wearables on the market but, rather, to pinpoint some timely, cost-effective, and feasible solutions for data collection pertaining to loneliness and social-isolation measurements. In Table 3, we show eight examples of wearables, their principle of operation, their typical cost, weight, and battery lifetime, their input interface, type (such as tag or Access Points (AP)), and body part where it is typically worn, the main measured parameters in connection to loneliness and/or social isolation, and the data types at the output (such as Comma Separated Values (CSV)), with the corresponding data extraction interface (e.g., local app or the manufacturer Cloud interface).

Examples of location monitoring devices for indoor and outdoor positioning are, e.g., the Pozyx system, comprised of an infrastructure of AP (e.g., installed on indoor walls) and lightweight tags (carried by users), and the MiniFinder Pico Global Positioning System (GPS)-based wearable device for outdoor measurements. The Pozyx system also has inbuilt accelerometers [99], which can provide information about the motion information of the users.

While Pozyx and MiniFinder Pico solutions are specifically intended for tracking the individual’s location and motion qualities, another metric that can be measured is the individual at rest. As previously discussed, sleep disturbance is a direct indicator of loneliness [100], and monitoring this particular metric could provide researchers with a deep understanding of the participants’ physiological and psychological state. Various smart rings, such as Oura [101] or Moodmetric, as well as many commercially available smartwatches, such as Smart Bracelet P11, are already capable of tracking the users’ sleep patterns or disturbances. These devices typically utilize PPG sensors to determine blood flow and the heart-rate analysis through ECG analysis. They can also monitor the user’s temperature or Electrodermal activity (EDA). Similar devices can monitor the pulse-oximetry levels or Peripheral capillary oxygen saturation (SpO2) levels of the user and procure data that could be utilized to determine the quality of sleep. Other sensor-collected data in the form of audio or light signals can also be used to provide additional information regarding the user’s surroundings [102], and such information can be utilized to determine whether the user is engaging in social activities, is sleeping, is alone, or is surrounded by people.

### 5.2. Sensor Obtrusiveness

For the sensor-system setup to capture and analyze human activities and be effective and lead to accurate outcomes, the sensors must meet an important requirement, wherein the user must have minimal or zero interaction with the system setup. In addition, especially when the focus is on older adults, the users should not be required to wear many hardware elements of the system; ideally, a device-free solution would be the best, but such solutions are still hard to be found in the current market. The ideal scenario for the participant’s relationship to the sensor device must be “to wear it and forget about it”, e.g., Pozyx-based accelerometer hardware setup to analyze human motion requiring four or more AP to be placed within a room [103]. On the one hand, such a relatively large amount of AP may cause discomfort to the user; on the other hand, too low a number of AP deployed within a certain indoor environment may adversely affect the quality of the collected data.

As multiple physiological metrics could be measured to analyze similar physical activities/actions, and numerous sensor technologies can be utilized to achieve that, it is important to consider the ease of use and the amount of needed user involvement when choosing a specific solution. For example, a simple solution with a single accelerometer sensor combined with a mobile phone, as depicted in [104], could analyze various human activities to a large extent. Furthermore, some systems could recognize activities via the use of a mobile phone [105], without any additional hardware required.

### 5.3. Energy Consumption

Another crucial element to consider in choosing a wearable solution for loneliness monitoring and management is the energy consumption of the implemented devices. Any hardware-based setup loses autonomy and requires more frequent maintenance from participants as the demand from a power source increases.

The energy consumption of the standalone/remote sensor hardware in the monitoring system is an important factor affecting the system’s autonomy. It is a critical aspect to consider, especially concerning the demography of the participants (i.e., older adults). If there is a constant requirement to charge or replace the energy source, the older participants experiencing dementia or other mental health issues may not be taking care of the sensors, leading to discrepancies in data collection. An example where older adults’ ECG metric were monitored with mobile phones can be found in [106]. The authors report that a frequent issue during the data-collection process was poor battery life. Today, smartwatches can perform a multitude of tasks with much more durable battery life than, e.g., conventional mobile phones. A major influence on the longevity of the battery is the communication mechanism of the sensor system. For example, short-range wireless devices utilizing WiFi or BLE typically consume less energy than devices that use long-range communication systems, i.e., 4G, 5G, etc. Exceptions here are the low-power wide-area solutions, such as Long Range Wide Area Network (LORAWAN) or Sigfox.

### 5.4. Data Extraction and Manipulation

The selection of the optimal sensors for the loneliness-management system also depends greatly on the nature of the data obtained and the protocols the sensors rely upon. The sensor systems should be capable of providing the data straightforwardly, through Application Programming Interface (API) or Software Development Kit (SDK). Having an easy-to-use Graphical User Interface (GUI) is also relevant. The monitored information must be available in raw form for further processing, where it could be analyzed with available training datasets and suitable ML or AI algorithms. Not all devices that monitor human physiological metrics, location or activity-monitoring metrics, or other loneliness-related metrics can extract the raw data. There is an important requirement to access raw signal values obtained from the sensor. Post-processing and data analysis relies on raw data, but, sometimes, accessing only the post-processed data may be enough. Such post-processing can be performed externally, as showcased, for example, in [99,107], wherein the obtained location data and ECG data were processed by an external server and a mobile phone application, respectively. Once the post-processed data is available on a Cloud server, it could be accessible through the Cloud or Ethernet/IP protocols. The post-processor can also have the ability to extract the features from the collected data and can subject it to the required classifier; this can be utilized to control the duration of transmission of the sensors to the post-processor hardware, thereby improving the efficiency of energy consumption [108]. More about connectivity solutions and Edge/Cloud computing is discussed in Section 7.1.

Another important aspect in the data extraction is the synchronization among the different boards/apparatuses used, for example, for the vital and ambiental parameters. In order to correlate different measures coming from different acquisition system, the timestamp of the acquisition is important. However, as the feeling of loneliness is unlikely to fluctuate with very fast granularity, synchronization errors of even several minutes may be still tolerable and may not affect the loneliness monitoring solution too much. The synchronization issues in the context of loneliness management and monitoring are not yet well studied in the existing literature. A good example of recent synchronization mechanisms for wearables can be found, for example, in [109]. The synchronization solution from [109] relies on a fractional-time concept, and it is able to achieve ultra low-power consumption. Time synchronization errors below 0.5 ms were achieved in [109].

### 5.5. Application Requirements

The biggest criterion affecting the selection of the sensor devices remains as the actual application or service for the planned implementation. Considering all the previously mentioned factors, the cost of the sensor system setup and the demography being studied are key additional factors that vary for each application. For example, if the focus would be on determining the perceived loneliness of older adults living either full-time in the assisted homes or living independently, but still visiting the assisted homes for socializing activities (e.g., gym, meeting lounge, arts and craft, weekly group activity, etc.), then, the main requirements regarding the sensor-based data collection would be to have (i) a lightweight and compact wearable; (ii) a long battery life; (iii) low or zero maintenance needed from the user; and (iv) an easy-to-use interface.

The utilization of smartwatches and mobile phones may require the participants to interact with a technology that may be too advanced or inaccessible to every member. The utilization of a ring sensor may also not be suitable for all, as the electro-dermal response of older adults may be inaccurate due to the aging of the skin. Smart rings also require frequent charging of the power source for continued performance. Therefore, one possible approach in the example above, to identify behavior patterns indicating loneliness or its onset, can be to monitor the location of the participants continuously while indoors (at the assisted homes, applying to all participants) and outdoors (applies only to participants who live independently, while away from the assisted home premises). The indoor location can be measured, e.g., with UWB sensors from the Pozyx system (which has a 30-month battery life on tags), and the outdoor locations can be measured using a GPS-based device, such as MiniFinder Pico. Both Pozyx tags and MiniFinder Pico devices are compact, have a very basic user interface that reduces difficulties for usage, and have a good battery life for their intended purpose. Based on the locations frequented and the movement patterns of the older adults, the level of loneliness or its onset can be identified through ML analysis of the collected data. Further examples are provided in Section 7.

Before going deeper into the ML-based approaches and wireless connectivity issues, an important aspect yet-to-be-addressed is the relationship between wireless technologies and architectural design practices and how the built environment can influence the perceived loneliness. These aspects are addressed next, in Section 6.

## 6. Relationship between Wireless Technologies and Architectural Design Practices

There is a gap between architectural and urban design and wireless communications industries. However, wireless communications have a central role within the smart buildings and smart cities’ paradigms, as well as a big and increasing impact on the operational energy of buildings [110]. The concept of ‘green buildings’ is one of the key aspects of building design paradigms of today.

Nonetheless, wireless networks are usually deployed when buildings have been already built [111], in contrast with other building systems impacting energy performance, such as lighting, heating and cooling, ventilation, and provision of water, which are considered at the early stages of the design process. The parts considered at the early stages of the design affect the building’s form, layout, and material choices from the beginning. Therefore, there is a need to develop instruments to guide the built-environment practitioners to predict and evaluate the wireless performance beforehand in their designs [111]. Specifically, regarding wireless systems used in loneliness and social-isolation interventions, it is highly relevant to design the forms and layouts of the older adults’ homes and environments according to energy-efficiency measures.

Various factors affect the energy consumption, such as: the building location (e.g., isolated or in a built-up area), the building layout, the room sizes, the aspect ratios, the raw building materials, the materials assembled, etc. Such parameters influence the received signal strengths during wireless connectivity; therefore, they affect the energy consumption on users’ wearables and mobile devices.

The absorption values of microwaves, as well as properties, such as conductivity, permittivity, and permeability, together with the shape and roughness of the building elements, vary substantially among different materials [112,113]. RF waves exhibit different behaviors, depending on the type of electromagnetic wave and on its frequency and wavelength [114]. Various researchers have elaborated on new building materials properties and configurations to ease wireless performance. For example, the authors in [115] studied how corrugated surfaces improved wireless-signal coverage, at selected frequencies, in the shadow region of buildings; the authors in [116] investigated how to overcome the attenuation of transmission through metal-coated glass (a kind of energy-saving glass, which is currently used extensively), proposing laying a frequency-selective surface on the coated side of glasses, for transmission enhancement. Considering this, older adults’ living environments, to incorporate AAL systems, including loneliness and social isolation interventions based on wireless technologies, should consider material choices for the system to work at its best performance.

The aforementioned reflections account for practicalities concerning the integration of wireless technologies in the built environment, specifically in older adults’ living environments. What follows refers to the opportunities offered and the challenges faced by combining the immaterial realm of data gathered through wireless systems with the physical body of architecture to alleviate, detect, and predict older people’s loneliness and social isolation.

Architect and thinker Juhani Pallasmaa [117] argues that: *“There is a constant interchange between our minds and our settings; as I enter a space, the space enters me. [...]. The experience of loneliness and isolation leaves the individual alone without an identification or interaction with the setting, whereas in the positive case of integration one feels accepted, supported and safe. Simply, we are fused with our settings and situations, and this unity supports the sense of self. [...] The fundamental task of art, architecture and urbanity is to mediate between ourselves and the world. This metaphysical and existential mediation has been the most essential task of architecture [...]. Architectural and urban spaces can either strengthen or weaken the sense of belonging, the meaningfulness of being, self-identity and self-esteem, which are all essential foundations of meaningful existence. [...] Simply distinct properties and qualities of physical and spatial settings give specific meanings to our sense of being, and make us feel participants, instead of outsiders or mere onlookers. [...] An urban setting or atmosphere can alienate and disconnect us from cultural, social and human context, or it can enroot us, and make us feel grounded, accepted and supported.”*

According to Pallasmaa’s words, good architecture by itself can enroot us, providing us with the feeling of being with, even if we are away from, other people, which can be regarded as the feeling of positive loneliness or solitude. Interestingly, from research on smart houses, the same kind of feelings have been detected due to a kind of “animism” of the house, when a certain intelligence is overlaid upon the aforementioned architectural qualities into the atmosphere of smart homes. As reported in [118], most of the older adult participants in their study missed the “voice” of “somebody” in their house when the smart system was uninstalled at the end of the experiment they participated in; through that presence, they felt cared about and less lonely.

If this is an instance of a successful alliance between architecture and new technologies for loneliness alleviation purposes, as mentioned before, in research, there are also pieces of evidence of AI systems that incorporate wireless technologies that can help to detect and predict loneliness and social isolation, based on older adults’ daily routines. However, from the perspective of the built environment’s implications in the accuracy of these technology systems, there are still challenges to be addressed.

To start with, future studies may benefit from greater considerations of contextual complexity surrounding the interventions. From an architectural perspective, the actual physical characteristics and conditions of the space where the experiments take place are not deeply considered, and they are facts that might be befuddling the results of the experiments. For example, as pointed out in [119], it is important to consider, among other indicators, urban planning variables, such as the proximity of resources and ease of transportation, as well as aspects that relate with the typology of the building where older adults live, such as if they are independent living units or a kind of community building. Other contextual characteristics might be acquainted, e.g., the accessibility barriers in the surrounding urban area of older adults’ houses, which might reduce their outings. If this is not considered, the predictive models might confuse this causing less mobility with that of the person being lonely, given that the systems positively correlate reduced outings with perceived loneliness and social isolation. Besides urban conditions, architectural features, including house layouts and spatial compartmentalization, spatial sharing possibilities, the physical and visual connection of interior and exterior spaces, and material qualities, might be considered, among other characteristics of the living context. In this sense, it is relevant that researchers find a way to implement the complexity and variety of living spaces in their experiments, if seeking accuracy in their results, and offer real and integrated solutions to detect and predict loneliness and social isolation.

Moreover, there are disciplinary frictions that arise when combining both big data engineering and architectural design practices. The detection and prediction studies mentioned are unidirectional, i.e., they infer loneliness and social isolation based on individuals’ behavioral patterns within certain environmental conditions, and not the other way around; they do not have the objective of analyzing the causes that generate the phenomenon for further intervention in their effect. For instance, if the system detects a diminution of outings of the older person, potential causes related to changing conditions in their environment might be considered and assessed before, or together with, the correlated psycho-social condition. This instance exemplifies a fundamental epistemological conflict that exists between outputs provided by the AI technologies and the way of producing knowledge of other practices, such as, in this case, architecture discipline. AI solutions provide data based on correlations, whereas architecture core practice act upon causal and reciprocity relationships, as well as through concepts. According to philosopher Byung-Chul Han [120], *“correlations are replacing causality. That’s-how-it-is stands for where How so? Once wavered [...]. Correlation represents a relation of probability, not of necessity. [...] Big data affords only extremely rudimentary knowledge, i.e., correlations in which nothing is comprehended. Big data lacks comprehension, and it lacks the concept [...]”*.

If we reduce the situation of adopting AI loneliness and social isolation detection and prediction systems to absurdity, we could imagine a dystopian scenario in which, for the systems to work, all the older adults’ houses and their context were to be laid out identically, so that it would fit with the parameters controlled by the AI system. Even though this is an extreme scenario, its sublimated vision raises an idea we should bear in mind: that these powerful techniques should be incorporated in comprehensive design processes and not as a substitution of them. They provide substantial data to be added to other complex and networked inputs that must be considered to intervene in the intricate phenomena of social isolation and loneliness.

All in all, it is clear that there is a need to bridge the existing gaps between disciplines, including smart technologies engineering, psycho-social, health, and architectural design ones, so, together, they can formulate integral interdisciplinary and site-specific solutions to help to “attune” [121] the environment with the “typical human situations” [122] and, thus, to contribute to increase the social inclusion of older adults and to alleviate feelings of loneliness.

## 7. Proposed Wearable-Based Monitoring and Management Solutions

A top-view block diagram of our proposed approach is depicted in Figure 4. This block diagram illustrates the generic conceptual framework without entering into the details of the sensors to be used, as the proposed architecture is valid for a variety of sensors. As it was explained previously, in Table 3, examples of sensors that can be integrated into the proposed solution can be positioning sensors, such as Pozyx-based systems and/or Pico MiniFinder, and stress detector sensors, such as Oura or Moodmetric rings, as well as other physiological sensors, such as sleep patterns, breathing rate, or heart rate. The conceptual framework relies on the assumption that data can be collected via various sensors, such as proximity sensors, accelerometers, etc., both with the help of the available infrastructure and in a collaborative D2D manner. The continuous or discrete data collected from these sensors is fed into an ML engine, where this data is pre-processed to extract various time domain, frequency domain features, behavioral and mobility information, social networking/check-in information, spatial information, and other location-related information from it. Note that any available data can be used as input to the ML algorithm. Some examples based on dummy data are later provided in Section 7.3. These extracted features are given as inputs to ML algorithms, such as SVM, Random Forest, NN, etc., to make predictions regarding the perceived loneliness among individuals and to offer recommendations toward alleviating or managing feelings of loneliness. ML approaches can be further used to classify the users according to pre-defined loneliness metrics for monitoring purposes and for helping in future spatial design (see further details in Section 7.5.2). Various features and information obtained from ML algorithms can also be used to identify behavioral and mobility patterns, which can be the basis of further socio-technological solutions (see other details in Section 7.5.1). Furthermore, the information obtained from the ML models can be used to create a loneliness monitoring and management ecosystem with a more advanced solution that could include recommendation systems, eHealth advice/support, social relationship improvement tools, etc. Such solutions are addressed in more detail in Section 7.3, Section 7.5.1, and Section 7.5.2.

The next two sections, Section 7.1 and Section 7.2, discuss the technical aspects of the proposed solution in more detail, in terms of wireless connectivity and ML algorithm choice, respectively. Then, a proof-of-concept based on dummy data is presented in Section 7.3. After that, Section 7.5.1 and Section 7.5.2 summarize different loneliness monitoring and management solutions.

### 7.1. D2D Versus Edge/Cloud-Computing Solutions

Today, competitive wearable devices have lightweight and common-sense sizes to offer comfort and benefit. Therefore, the priority, from the design point of view, is given to sensors to provide the basic task of the device while limiting the space for the Central processing unit (CPU). To overcome this issue, developers are commonly starting to utilize the concept of computational offloading in order to reduce the processing load on the end-device [123]. By this, a wearable device obtains an opportunity to delegate the collected data/task to another more powerful unit. As such offloading naturally comes as a trade-off to the introduced transmission overheads, researchers in the wireless domain are interested in finding energy-efficient ways of communication that enable continuous connectivity and sharing of data [9].

The comparison of the major communication-related qualitative features is provided in Table 4. The purpose of this table is to showcase examples of existing technologies which can enable wireless connectivity in a D2D, Edge, or Cloud mode for the survey’s perspective. However, our goal is not to pinpoint the exact wireless technology suitable for the application at hand. The main purpose of Table 4 is to give a short survey of the main available wireless technologies and their fit to various architectures, such as D2D, Edge, or Cloud computing Indeed, significantly, different technologies may have tremendous differences in terms of throughputs and delays, thus making the selection of the application to be executed over those a challenging task. From Table 3 and Table 4, it is clear that the vast majority of sensor solutions rely on BLE wireless connectivity with the mobile device.

The parameters discussed in Table 4 include the frequency bands where a system operates (as such, frequency bands also influence the user-to-user distance, which can be covered in a D2D or Peer-to-Peer (P2P) mode or the coverage of a certain infrastructure), the allocated channel bandwidths (typically, higher bandwidths can support higher throughput and, thus, more complex services, but they may also require more expensive user devices), the maximum supported data rates (or throughputs), and the achievable end-to-end latency and reliability levels, which are measures of the quality of service from the point of view of the wireless technology [124], such as Wireless Local Area Network (WLAN) or cellular 4G/5G connectivity.

Overall, most of the technologies allow for flexible selection, ranging from the physical layer to Medium Access Control (MAC) layer, depending on the application. Here, the major factor separating Bluetooth- and Institute of Electrical and Electronics Engineers (IEEE)-based wireless connectivity from Third Generation Partnership Project (3GPP) cellular systems is related to the frequency use. The first group operates in the unlicensed Industrial, Scientific, and Medical band (ISM) bands, where no guaranties of Quality of Service (QoS) could be provided, while the latter group of cellular technology operates in licensed bands and puts the pressure on the shoulders of the telecommunications giants. Evidently, as the interference level in the unlicensed ISM bands is higher than in the licensed bands, ISM-based technologies typically provide lower data rates than cellular-based technologies. At the same type, the ISM-based technologies are typically meant for higher energy efficiency than the cellular ones, as they operate at a shorter range, making the technology selection even more application-specific. Table 4 also specifies the Release (Rel.) numbers pertaining to 3GPP standardization.

Promising enablers to achieve the communications for the offloading scenario are computing in: (i) a remote Cloud, (ii) a close-by network periphery (i.e., Edge computing), or (iii) a D2D mode. Conventional Cloud computing is a paradigm where the data is transmitted via Wide Area Network (WAN) (e.g., 4G/Long Term Evolution (LTE), or 5G) to the resource-rich server for further processing or storage. At the same time, the mobile device connects through wireless mobile communications and unloads for other tasks [132], i.e., to a robust data center or other infrastructure operating with big data volumes. The number of data centers (or Cloud servers) is typically limited, while the demand for remote services is higher and higher. Therefore, the idea to compute tasks that are not highly intensive in the network’s periphery (or Edge) has become significantly more attractive for the network operators. While Cloud computing is already present on most devices, by default, Figure 5 illustrates that the data may not necessarily be sent to a remote serve but, rather, be delivered to the closest less power-dependent node via D2D or Edge paradigms [133]. Migrating the computational tasks to the Edge through WLAN (e.g., WiFi) provides fast computation with lower latency as it is physically located closer to the user. Edge servers support delay-sensitive applications and provide real-time services with low latency due to avoiding additional handover on base stations [134].

Finally, the D2D paradigm does not rely on any additional infrastructure, such as data centers or a network’s periphery devices, but enables direct connectivity between user devices. In such a way, the devices of two users found in the proximity of each other can exchange relevant information and enable various services, such as proximity-based social networking, etc.

Notably, Edge and D2D computing require a certain level of awareness and management from the infrastructure network to solve high mobility related to personal handheld and “carriable” devices [135]. In contrast, the Cloud paradigm relies on any level of infrastructure connectivity, which may result in significantly higher delays compared to D2D. D2D connectivity could be executed in a P2P manner without sending bulky data through any long-range multi-hop links.

To sum up, D2D is a communication paradigm that could be defined as direct communication between mobile devices that provide better spectrum efficiency and ultra-low latency if compared to cellular networks [136]. It is a state-of-the-art communication paradigm, which is already standardized and available in recent LTE releases, for proximity-based data sharing services based on such technologies as WiFi-Direct, BLE, LTE direct, and those, in turn, could be split into in-(licensed)-band and out-band ones [137]. In-band D2D uses the licensed spectrum, while the out-band operates in the unlicensed spectrum, shared with other ISM-band applications. Despite the advantage of typical higher user privacy than communication through the base station, the D2D architecture has the main drawback of an increased risk of inter-user interference, e.g., due to the interference between cellular communications and in-band D2D links or between other wireless devices using ISM bands and the out-band D2D links.

Overall, D2D is developed to offload cellular networks, from a bandwidth reuse perspective, to improve system performance and QoS [138], while computational offloading is considered as one of the promising technologies to utilize the D2D potential. One of the recently-emerged computing paradigms for wearables and smartphones based on D2D is the so-called *Dew computing*, which is an ad-hoc distributed system deployed for processing tasks locally [139]. The Dew paradigm is expected to provide real-time operations, on-demand, with reduced communication costs. The ad-hoc network could exploit the devices near and offers significant computational power. Dew computing can be seen as the hybridization between the other discussed computing paradigms and could find a yet-not-investigated potential in the context of AAL and loneliness management ICT solutions.

As a conclusion of the discussions in this sub-section, the wireless-communications enablers of today already allow for efficient computational offloading over wireless links, making the development of loneliness mitigation strategies one step closer to being implemented on personal devices of older adults. To specify the wireless technology, BLE is a recommended connectivity solution for sensor devices as it benefits in terms of energy consumption, privacy, and latency. In particular, Dew and D2D computing paradigms are promising paradigms that also deserve further investigation in the context of various services for older adults, such as services to monitor and manage loneliness and social isolation.

### 7.2. Machine-Learning Aspects and Recommendation Systems

Already today, ML solutions can predict the loneliness prevailing among older people. It has been observed from the previous studies, e.g., in [47], that the datasets required to estimate or predict loneliness levels are typically multi-modal data, consisting of behavioral data, mobility data, health-related information, social interaction data, location-based data, etc. Data related to health, mobility, social-interaction levels, and user positions can be obtained from different sensors, such as heart rate sensors, sleep monitoring sensors, or position-related sensors. More details about such sensors are provided in Section 5. However, behavioral data, such as anxiety level, depression level, stress levels, etc., are typically obtained from questionnaires, even if stress-estimating sensors, such as Moodmetric rings (see Table 3), already exist on the market.

The data collected from sensors are mostly continuous, with integer or floating-point data types. In contrast, the data obtained from questionnaires are categorical values, based on scales, such as the Likert scale (as seen in the typical loneliness metrics discussed in Section 4.1). Various ML algorithms can be used to analyze this variety of data. The data needs to first be pre-processed to extract the relevant features, such as time-domain and frequency-domain values, and then it can be fed into ML models to perform classification and prediction tasks or to serve as a basis for future recommendation systems.

In terms of what kind of ML algorithms are best suitable to work with such data, a systematic literature study conducted in [140] for ML algorithms used for eHealth applications showed the results depicted in the pie chart in Figure 6. These results from Figure 6 are based on a systematic review of 67 scientific articles. It can be observed from the pie chart in Figure 6 that NN algorithms, SVM, and Random Forest are the most encountered ML algorithms in dealing with user data for eHealth applications (e.g., they were used in 28%, 27%, and 19% of studies, respectively). These three algorithm classes were studied to have better classification performance for analyzing the continuous sensor data over other algorithms. NN algorithms used for analyzing sensor data include ANN or shallow neural networks, Deep Neural Networks (DNN), multilayer perceptrons, and Long Short-Term Memory (LSTM) networks. Other NN-based algorithms, such as Convolutional Neural Network (CNN) or Recurrent Neural Network (RNN), were mainly used for image classification but not widely encountered in the context of eHealth data. DNN and multilayer perceptron showed better performance for classification tasks for time domain and frequency domain values than other studied algorithms in [140] and references therein. Ensemble learning techniques have also been used in 11% of studies. These techniques combine the prediction from multiple models to obtain better predictive performance. Ensemble learning methods include techniques, such as bagging and boosting methods. Bagging methods combine the predictive performance of several decision trees executed on different samples, whereas boosting methods work by correcting the predictions made by other models and output a weighted average of multiple predictions. Other classification algorithms, such as Bayesian methods and kNN, were used for comparative analysis. To study our hypothesis of predicting loneliness levels using ML algorithms, we used three ML algorithms, Logistic Regression, Random Forest, and SVM, utilizing dummy data represented the categorical and continuous data. All three algorithms showed good accuracy ranging from 97% and 99%.

Table 5 compares various ML algorithms in terms of their advantages and disadvantages.

From the discussion, it can be inferred that, for analyzing the features from continuous sensor data, NN, SVM, and tree-based algorithms, such as Random Forest, show better performance than other ML algorithms; thus, we are recommending one of these three algorithm types (namely SVM, NN, or Random Forest) to be used for loneliness monitoring and prediction. A proof-of-concept to validate this recommendation of monitoring and predicting loneliness is shown next, in Section 7.3. It uses three machine learning algorithms: Logistic Regression, Random Forest, and SVM, based on dummy data. In scenarios with real-time data, different sensor data, data types, and algorithms (mentioned in Table 5) could be used for analysis.

### 7.3. Recommendation Systems for Loneliness Monitoring and Prediction—A Simple Proof-of-Concept

Our hypothesis to be studied here via a simple proof-of-concept with dummy data is that ML models for loneliness predictions can identify the relationship between sensor-collected data and user-defined perceived loneliness data. Such models can not only be the basis of loneliness monitoring and management ecosystems, as shown in Figure 4, but it can also function as recommender systems or decision-support systems to improve the social relationship of older adults. Collaborative filtering approaches for developing recommendation systems can build a user-tailored model using the user’s past behavior. In the context of loneliness prediction, the recommendation system models can help predict the loneliness state of other individuals. Moreover, such models can also be used for continuously monitoring the person’s loneliness state. If spatial information (see Section 7.5.2) is also available, the ecosystem can include also cross-programming elements, as discussed later in Section 7.5.2.

Our proof-of-concept study, detailed in what follows, aims at predicting the loneliness level of older adults based on multi-modal sensor data and five loneliness indices. In the absence of real-life data (as the open-access data from Figure 2 and Figure 3 is lacking temporal multi-sensor parameters, and it only provides a single index per user), we have generated dummy data to prove our conceptual approach. For this purpose, we used randomly generated categorical and continuous data. We selected both categorical and continuous data because, in real-time scenarios, categorical features indicate values from questionnaires or personality traits. In contrast, continuous data indicate values collected from sensors, such as mobility data, health-related data, stress and anxiety levels, etc. Our randomly-generated dummy dataset contains values for 1000 instances, indicating 1000 users, and 5 metrics, indicating answers to the questionnaires regarding behavioral or personality traits, as well as sensor data. We used floating-point values for 5 variables, indicating values from sensors, at a particular time. Each metric or sensor value was weighed between 0 and 1. The value of weights was randomly generated and normalized. The loneliness index was then calculated based on the weighted sum of all the metric/sensor values, as mentioned in Equation (Equation 1). These values, combined in a loneliness index, generated the labels for the training and testing data for ML algorithms. To assign the proper labels, we used the distribution of the loneliness index, as shown in Figure 7.

In our example, as seen in Figure 7, the distribution plot follows the Gaussian distribution. However, it is to be noted that, with real-time sensor data, the distribution may not follow the Gaussian distribution; therefore, the expected classification results with real data may be poorer than what we report in this example. To label the values, we used the mean value of the distribution as a threshold. For simplicity, we only used two labels in our model, 1, indicating high loneliness (values above the threshold), and 0, indicating low loneliness (values below the threshold); clearly, multi-level labeling with more than two levels is also possible. For the ML part, we selected the three algorithms which were most frequently used to analyze the sensor data according to Figure 6. This conceptual model used Logistic Regression, Random Forest, and SVM. SVM and Random Forest were selected, based on the discussion at the end of the previous sub-section. We did not use NN algorithms as these algorithms typically require a good number of samples, and, for our conceptual model, we used only 1000 instances, for a moderate complexity. The logistic-regression algorithm was also included for comparative purposes to show that it has lower performance than the previously recommended ones (SVM and Random Forest). The dummy dataset was divided into 80% training data and 20% testing data. The evaluation criteria used were:Accuracy—This metric tells the correctly predicted observations (positive or negative), divided by the total number of observations. A positive observation here means that a user in the high-loneliness class is correctly predicted as a lonely user.Precision—This is defined as the correctly predicted positive values divided by the total predicted positive values.Sensitivity—This is defined as the proportion of actual positives identified correctly among all positive and negative predictions.ROC-AUC—The Receiver operating characteristic curve (ROC)-Area under the ROC curve (AUC) metric tells how well the model predicts the classes (here, two classes: high loneliness versus low loneliness). Its value lies between 0 and 1. A value close to 1 indicates a better model than a lower value; for the two-class prediction, a value close to 0.5 indicates a random model.

The results based on the three algorithms mentioned above (Logistic Regression, Random Forest, and SVM) are shown in Table 6. All these algorithms show very high prediction levels in terms of accuracy, sensitivity, and precision (values between 97% to 99%), and the ROC-AUC curve value was always 1, indicating models with good prediction capabilities. The best performance in all the metrics was shown by Random Forest, followed by svm. The excellent performance can be explained to some extent by the fact that we use dummy data; however, this simple proof-of-concept also shows that our conceptual model based on ML and sensor-based and behavioral data can be used to predict loneliness levels of older adults.

### 7.4. Loneliness Monitoring Solutions

The social importance of studying loneliness is directly related to improving the quality of life and reducing mental disorders and mortality [4]. If we are talking about older adults, classical methods of dealing with loneliness may not solve the problem since loneliness in older adults is more related to health and social status than for younger adults (recall that the central factors of loneliness discussed above, in Section 3.2). Implementation of the digital environment and modern technologies can help older adults to work against mental-related problems.

Monitoring mental issues, such as loneliness, is typically a non-trivial study for researchers. One of the possible solutions that we are using in this work is to collect multi-modal data from a variety of sensors and analyze it to extract relevant features and inter-dependencies to derive user-tailored solutions.

Our proposed architecture envisions that, in order to improve the social relationships and to decrease feelings of loneliness in the elderly, some of the key technological components of a loneliness-monitoring service are: (i) the on-demand and ubiquitous positioning and tracking of a person’s movements, in order to detect the daily activities, social interactions, and changes in regular patterns, and to promote social networking with nearby individuals; (ii) powerful ML algorithms for designing user-tailored and system-optimized recommendation systems; (iii) the use of reliable, low-cost, and long-battery life interconnected wearable sensors (and, possibly, social robots), relying, for example, on energy harvested from surroundings and intelligent energy optimization methods; and (iv) the innocuous and secure interaction with sensors and, possibly, with various social robots in order to ease the daily tasks and movements, to ensure a first-aid fast response, and to have daily incentives for increased socialization and for a healthier lifestyle.

As emphasized in Section 5, the data collected from sensors can typically be divided into two main groups:Geospatial data: This first group consists of geo-tagged data, such as mobility data [6] or any location data. Devices with built-in geo-positioning can provide the route patterns and person’s location and proximity to other people. Based on this information, it becomes possible to identify a person’s most popular space, favorite one, how much time they spend there, and whether they are active enough. In addition, proximity information can infer social networking activities concerning other people.Socio-medical data: The second group consists of social and/or medical-related data collected from wireless devices [144]. The modern technological market offers a lot of various devices for monitoring sleep, ECG, anxiety levels, stress levels, etc. Examples of such sensors are discussed in Section 5.

The data collected from sensors can be transmitted to a computing-rich device for further processing. The showcased ML algorithms can analyze patterns, detect deviations, and send back the results to the users or their families or caretakers [145].

### 7.5. Loneliness Management Solutions

This sub-section expands the loneliness monitoring solutions addressed in the previous sub-section with more ideas toward managing and alleviating loneliness. We will address the loneliness-management solutions from two joint perspectives: the socio-technical perspective, taking into account social psychology and technological aspects, and the spatial perspective, taking into account the architectural or building environment and the gerontology aspects. For the sake of the completeness of this survey, both technological-based and non-technological-based solutions are described, with the reminder that the proposed solution of wearable-based ML algorithms falls into the first category, of socio-technological solutions from Section 7.5.1. Nevertheless, an overview of other complementary solutions is included in Section 7.5.2. This overview of various, technology-based, and non-technology-based solutions is also necessary to point out that effective solutions to alleviate loneliness need to take into account the multidisciplinarity of the problem; therefore, cross-disciplinary answers are likely to reach better results than methods relying on single dimensions.

#### 7.5.1. Socio-Technological Solutions

Interventions aiming at alleviating loneliness have become a prominent part of the research literature [146,147], including technology-assisted interventions [19,62,148].

Such technology-assisted interventions can use a variety of ICT tools, ranging from low-tech to high-tech solutions. However, knowledge of the exact mechanisms leading to reduced loneliness is still limited. A systematic literature review in [19] found that, for physical technologies, such as social robots, the physical presence and interaction capabilities of technology may be relevant features in loneliness reduction that could be addressed in more detail from a human perspective in the future. It is clear that older adults’ perceptions of the use of technologies vary, including both advantages and drawbacks [72], and the role of technology should be supportive, rather than dominant, in combating loneliness among older adults. However, technology has a lot to offer in the management of loneliness and social isolation [149]. The major goal is to establish and enhance seamless communication and connectivity among the elderly and their community. This is very important, as it promotes collaboration and interaction between the elderly, their family, and caregivers, enhancing their health and social interaction.

The application of technology includes the use of advanced communication technologies (telecommunication, Augmented Reality (AR)/Virtual Reality (VR) applications, social robots, and video-conferencing apps) to alleviate loneliness and improve social interaction [150,151]. It also extends to the use of age-friendly apps and digital games [152,153], smart mobility tools [154], and integrated open platforms [155,156,157,158] utilizing AI, IoT, wearables, etc.

The possibilities of VR to alleviate loneliness and social isolation have been studied in [159,160] and the video-conference interaction program in [161]. The results of these studies were positive, but engaging family members may partly explain the decline in loneliness and depression to the programs, instead of other more distant social contacts or strangers. The online friendship enrichment program in [162] showed a decline in overall loneliness but not in daily loneliness. Diverse online interventions are shown to affect the social and mental well-being of older people, but the results call for studies with different population groups and looking for long-term effects of the interventions to develop tailored programs with lasting effects. Online interventions to reduce loneliness and social isolation offer one tool in addition to other face-to-face individual and group interventions.

The study in [151] demonstrates how advanced communication technologies were applied to reduce loneliness and to improve the quality of life in the elderly. The study was based on the use of the Uniper-Care technology, which was developed to facilitate social connectivity due to its effect on the well-being of elderly persons. Through the installation of the Uniper-Care technology on the TV sets of the elderly persons, the older adults were able to participate in social activities remotely and interact with their friends and family via video calls. The study was carried out for about five weeks, and the results showed a reduction in loneliness and depression based on the participants’ emotional well-being data gathered before and after the study, using standards, such as UCLA Loneliness Scale or Patient Health Questionnaire (PHQ9) Depression Screener.

The use of smart-mobility tools also helps to alleviate loneliness and social isolation. These tools (basically mobile applications) are designed to assist the elderly with any mobility problem they may be facing. It involves the specification of the mobility routes of the recommended social places and other places of interest, such as parks, recreational centers, malls, etc., frequently visited by older adults. It provides an easy modality for the elderly to visit and connect with their friends, family, and community. It also provides recommendations on possible social places to interact with people. A typical implementation by the Capital Regional District (CRD) of Victoria is provided in [154].

The use of digital games and age-friendly applications helps reduce loneliness and social isolation in the elderly. It has even become more prevalent during this period of the COVID-19 pandemic as the mobility of the elderly has been reduced, voluntarily or decree-based, to diminish the likelihood of getting infected. Elderly persons can use age-friendly applications and play digital games to connect and collaborate with their friends and family. Age-friendly apps [163] come with icons and fonts that are easy to read and understand, thus making navigation easy. In addition, digital games also aid the cognitive development of the elderly and help to improve their social skills, as they become more socially inclined [152,153,164].

In addition to the different sets of technologies mentioned in the foregoing paragraphs, the European Commission is also promoting several initiatives for managing loneliness, social isolation, health care, and improving the quality of life of the elderly through the development of integrated open platforms, which utilize technologies, such as ubiquitous computing, big data, semantic web, Cyber-physical systems (CPS), IoT, Human-Computer Interactions (HCI), etc. These initiatives include several EU-funded research projects, such as ALFRED [155], where a solution has been developed to address social inclusion, personalized care using sensors, and cognitive impairments prevention using gaming. In the AMICARE project [156], the activities of the elderly are monitored and evaluated using a combination of sensors embedded in their furniture. In the PACO project [157], a healthy dietary habit is encouraged in the elderly, while also concurrently reducing instances of loneliness. Finally, the PHARAON project [158], which is still ongoing, aims to foster smart and active living in the elderly through the development of interoperable platforms with advanced analytics and smart wearables.

In addition to the ML-based monitoring by analyzing data from sensors, more complex solutions based on so-called *social robots* or *robotic pets* also exist today. For example, in [165], the authors used a social robot (a robotic seal) called Paro (Personal Assistive Robot) to detect depression and loneliness and the relationship between them. Paro robot was equipped with tactile sensors to detect light, sound, and touch. The elderly were given Paro robots to interact with for 24 h, for a week, for the study. The measurements were collected in two stages—before and after Paro’s intervention—from the generally accepted questionnaires from global organizations, e.g., the Geriatric Depression Scale Short Form (GDS-SF), the UCLA Version 3, and the World Health Organization Quality of Life Questionnaire for older adults (WHO-QOL-OLD). The positive psychological effects after interacting with Paro and its capability to improve loneliness among older adults were also discussed earlier in [166]. Paro reacts to its name and has vivid expressions that have been argued to enhance social interaction and psychological well-being, as well as decrease depression and loneliness, of older adults [165,167].

In addition, barking and meowing animal figures with sensors have been shown to have positive effects, especially among those adults who have few opportunities for social connections [168]. The positive effects need to be put into context since the results have varied according to the study population, social and educational resources, residential context, and the length of the intervention. In the study of Hudson et al. [168], the participation rate was low, with a large number of people (n=3660) refusing to participate; moreover, 86% of those who participated (n=277) had previously owned a pet. The study in [169] showed that dogs and interactive robot seals do have some positive effects on depression in nursing-home residents by increasing communication and tactile stimulation. Their study showed that physical presence and an opportunity for a certain level of interaction are important features of both real animals and robotic pets.

However, to ensure the smooth adoption and full implementation of the solutions described in the previous paragraphs, adequate training must be provided to the elderly on the use of the different technologies as this is essential to be able to achieve the goals of reduction of loneliness and social isolation and to promote social connectivity and well-being in the elderly.

#### 7.5.2. Spatial and Other Non-Technology-Based Solutions

Spatial solutions, such as shared spaces and common spaces, whether in private or public premises, outdoors or indoors, which are properly designed and programmed with adequate activities and with adequate smart technology infrastructure, could help older adults to maintain the two conditions mentioned above and, thus, help them to improve their quality of life, including the avoidance of social isolation and overcoming of perceived loneliness.

The social value of common and shared spaces depends both on design and programmatic (i.e., activities scheduled to take place in a certain space) approaches [170], which the implementation of technology can enhance. As an urban planner and thinker, Jan Gehl [171] suggests that *“architects and planners can affect the possibilities for meeting, seeing, and hearing other people”*. Thus, visual, acoustic, and thermal comfort and performance, material choices, and layouts, ensuring degrees of privacy and gradual progressions from the intimate realm to the public one [172], as well as procuring easy ways to navigate and clear transitions between areas, are crucial design factors to take care of for the success of convivial spaces. Regarding the programmatic issues, besides the logical adequacy of the activities to the age-group and spatial setting, the compatibility between potential simultaneous doings, among other obvious problems, could also be the key to success. In particular, when it comes to social interaction, this compatibility-based approach needs to incorporate also the possibility for the unexpected to happen to ease non-conventional relationships. The architect and theorist Bernard Tschumi [173] suggested interesting programmatic strategies in this regard, which can also be applied to the kind of spaces we are focusing on. It is what he calls *cross-programming*, i.e., an unexpected combination of programs and spaces. It would be the case of spaces for inter-generational integration to avoid ageism and age-segregation problems. Current research reveals a need for architectural solutions for the social inclusion of older adults as researchers have identified multiple benefits of inter-generational integration [174]. Cases of success are, for instance, the Yoro Shisetsu premises in Japan where children and older adults are mixed, or, for example, the *Dutch multi-generational housing model* of exchange of volunteering time of young people to help older adults for free accommodation.

Cross-programming works well also at the urban scale, for instance, to avoid cases of segregation of disadvantaged people and minorities in early stages of life, so there are no ghettos that would create problems in later-life to age-in-place, as the case studied in [55], already mentioned above. In this regard, an example of good practice is the Hunziker Areal neighborhood in Zürich, where a share of living units for minorities is reserved to integrate people at risk of exclusion within a lifelong community. The living arrangements in this neighborhood are significantly varied, including co-housing and co-living arrangements, as well as a series of spaces to socialize in various manners. The offer is so rich that people can stay in the neighborhood even if they have to change homes. Remaining in the same district implies planification of a variety of housing choices and facilities in it [170]. This concept is associated with the lifetime-homes concept, which includes the key architectural areas of usability, adaptability, accessibility, inclusion, and lifetime value of the living premises, including living units and common spaces.

Reducing loneliness and social isolation of older people has also been a target of numerous non-technical intervention studies, which shows the weight given to this topic by several reviews [36,175,176,177]. Such non-technology-based interventions represent a wide variety of programs of psychological therapies and social support (discussions, counseling, therapy, educational programs, and training skills), social activities, physical activities (fitness programs, recreational activities), arts-based programs, horticultural activities, and inter-generational programs [36,175,176,177]. The strongest effect in decreasing loneliness and social isolation of older people, according to [177], has art-based community programs, horticultural interventions, and new technology-based interventions. The results of an earlier review in [36], on community-dwelling people, had similar findings, supporting the role of educational- and skill-building interventions. Such interventions can also support the social-network maintenance and enhancement, thus leading to behavioral changes.

The role of new technology in interventions has multiplied only during the past decade, which likely explains the difference in the results in this regard. Non-technological interventions for reducing loneliness or strengthening social relationships can be based on theory-based approaches [178] or more individual approaches, such as mindfulness-based stress reduction [179].

Interventions, such as the *‘tai chi qigong’* program, which combined community-based and individual approaches, were presented as safe and feasible interventions to improve social networks among elderly [180].

The recent review in [175] emphasized the meaning of community-level interventions and the positive effect of having the interventions based on existing social structures and services to give them continuity. The authors in [175] stated that, to decrease the loneliness and social isolation of older people, the intervention programs need to support the autonomy of older people while also supporting the creation of new social contacts and fostering a sense of belonging to the community. While the study in [175] referred to non-technology-based interventions, their conclusions are likely to also be valid in the context of technology-based interventions.

A question that needs to be raised in future studies and interventions is whether the needs and well-being of older community-dwelling people differ from those living in residential care.

According to [176], horticultural therapy, reminiscence therapy, and laughter therapy were the most effective therapies in decreasing loneliness among older people living in long-term care facilities. Interventions addressing human-animal interaction and pet-assisted therapies, particularly in residential care and assisted-living facilities, have garnered huge interest during the last decade [181,182]. Some intervention programs have compared interaction with real animals versus robotic pets. While there was no clear difference between interventions with real and virtual pets in decreasing loneliness and increasing well-being, older people continued their interaction with real dogs longer than with robotic pets [169,183]. The visits of volunteers with a dog versus those without a dog had more effect on wellness and overall social activity and health of older people [169]. The study in [184] with community-dwelling people showed some support for the effect of pet therapy on the physical health and social wellness of older people. According to several studies [181,182,185], accumulating evidence shows that animal-assisted interventions have the potential to increase social and mental well-being and decrease loneliness and social isolation of older people by offering companionship, as well as by improving their social and physical activities outside the home environment. However, the results of the intervention studies are heterogeneous and often ambiguous due to small sample sizes, differences in protocols and reporting results, and different populations groups. Several studies [65,178,180] have stated that the interventions could not straightforwardly state that they had lasting effects, even if positive results are reported during or after the intervention. In randomized control trials, the key factor to be considered when weighing the meaning of the results is the similarity of the control group (e.g., the studies in [178,179]). Interventions may focus on community-level or individual-level strategies or a combination of these. The social resources, biographical elements, individual aspirations, residential context, and the participant’s health status and functional ability are taken into account in different ways. To summarize, the key message of the studies quoted here is that interventions seeking to tackle loneliness and social isolation of older people need to consider community, social, and individual contexts and address diverse populations and residential contexts.

In conclusion, in addition to technological solutions, planning and architectural solutions, such as the ones described in this sub-section, together with non-technological healthcare and social interventions and policies, all aided by smart technologies systems, could comprise integral responses to provide social support and maintain the physical competence of older adults, whether living independently or in institutions. While homes and senior facilities are the central space for people to grow older, there is also a whole system of out-of-home environments, both indoor (common and shared spaces, workplaces, shops, cultural and sports centers, health and care facilities) and outdoor ones (parks, streets, sports fields) that are important arenas for activities and socialization that could help to overcome social isolation and perceived loneliness. As suggested by various research sources, including the Royal Institute of British Architects [186], there is little research in out-of-home spaces, or on non-residential buildings (other than healthcare buildings), for older adults, especially regarding retail, leisure, civic, and workplaces. Thus, there is room for collaborative research between engineers, social scientists, and architectural and urban designers to pursue smart age-friendly living environments to ease older adults’ lives, including overcoming social isolation and perceived loneliness.

## 8. Conclusions and Future Perspectives

Promoting functional capacity, enabling social networking to avoid isolation, and preventing feelings of loneliness are the real keys to success for actively aging populations. Learning behaviors and patterns from spatio-temporal data collected from wearable sensor equipment represent a rich research field offering real opportunities, especially if recently evolving ML algorithms are to be exploited.

Our paper offers a multidisciplinary survey of loneliness management solutions, with a particular focus on wearable-based sensor-collected data processing solutions for monitoring and/or alleviating perceived loneliness and social isolation. The technological solution took into account also the constraints coming from the socio-psychological, gerontological, and built environment domains. A generic framework was proposed with in-depth details about possible sensors to collect data, possible wireless connectivity solutions, and possible ML algorithms. The generic framework was further narrowed down to a case study with dummy data collected from various continuous-valued sensors and three selected ML algorithms. However, the overall framework is not limited to only the considered three ML algorithms nor to a particular sensor type, but it can be adopted on a wider scale, according to the available sensors. Positioning sensors, such as UWB-based Pozyx or GNSS-based MiniFinder Pico, together with physiological sensors, such as Oura or Moodmetric rings, are good examples of sensors that can be used in the context of loneliness monitoring and management.

Overall, the identified challenges could be classified based on the Venn diagram from Section 3, i.e., into technological (T), socio-psychological (S), architectural (A), and gerontological (G) ones. The main ones identified are given in Table 7.

Technology challenges include the computing paradigm, wireless connectivity, and security and privacy preservation, all together forming a sophisticated ecosystem, which is coupled with the problems of a highly distributed environment. Notable is that its operating also brings architectural challenges referring to the living modalities; for example, the aging-in-place concept has many benefits, but, depending on the location and neighbors, it can also create several problems, as discussed in Section 3 and Section 7.5.2.

Social challenges include, for example, the subjective interpretations of loneliness and social isolation and the complex and multi-faceted dimensions of loneliness. In terms of gerontological challenges, the issue of dealing with physical or cognitive impairments at older ages represents two of the most important and hardest-to-address issues.

Socio-technological challenges comprise, for example, the mechanism of choice of a viable loneliness metric, among the several existing ones. As seen in Section 4, there are validated and reliable subjective metrics for both loneliness and social isolation, such as the UCLA and the DJGLS scales for loneliness and LSNS for social isolation. Validated subjective metrics result from long-term development work, and there are many versions of these scales. The choice of the measuring instrument to be used is never a neutral decision. Therefore, scholars should always pay close attention to which measure suits their study objectives and target population best to produce valid and reliable results.

Other inter-disciplinary challenges are, for example, the trade-off between various resources and social sustainability (e.g., a high amount of single-dwelling houses may have a negative impact on the long-term sustainability of resources) and how to attain a close-to-zero barrier in adopting new ICT approaches and devices for an older generation.

Recommended solutions to address some of the challenges mentioned above are tackling the loneliness and social isolation issues from a multidisciplinary perspective, such as combining technological, social psychology, gerontology, and architecture views and striving to adopt holistic user-tailored approaches.

In terms of technology, hybrid solutions combining the use of low-cost, low-power IoT sensors and wearables at the user side with the use of D2D connectivity and Edge/Cloud-based intelligent data storage and processing are also to be envisaged. In addition, ML algorithms are likely to play important roles in loneliness monitoring and management eco-systems in the future.

One of the envisaged solutions is to attain low levels of loneliness through aging-in-place, cross-programming, and the creation of social networking hubs with broad coverage and low power consumption. In addition, combining technology-based with spatial and other non-technology-based interventions is likely to give better solutions than focusing on technology-based solutions alone. Nevertheless, new solutions must also be found to achieve close-to-zero learning barriers and high acceptability of technology among older adults, and these remain topics of future research.

## List of Acronyms


**3GPP**

Third Generation Partnership Project
**4G**

Fourth generation of cellular networks
**5G**

Fifth generation of cellular networks
**AAL**

Ambient-assisted living
**AI**

Artificial Intelligence
**ANN**

Artificial Neural Networks
**AP**

Access Points
**API**

Application Programming Interface
**AR**

Augmented Reality
**AUC**

Area under the ROC curve
**BLE**

Bluetooth Low Energy
**CPS**

Cyber-physical systems
**CNN**

Convolutional Neural Network
**COVID-19**

Coronavirus Disease 2019
**CPU**

Central processing unit
**CRD**

Capital Regional District
**CSV**

Comma Separated Values
**D2D**

Device-to-Device
**DJGLS**

Dong Jong Gierveld Loneliness Scale
**DNN**

Deep Neural Networks
**ECG**

Electrocardiography
**EDA**

Electrodermal activity
**EEG**

Electroencephalography
**EIPs**

European Innovation Partnerships
**GNSS**

Global Navigation Satellite Systems
**GPS**

Global Positioning System
**GUI**

Graphical User Interface
**HCI**

Human-Computer Interactions
**ICT**

Information and Communications Technology
**IEEE**

Institute of Electrical and Electronics Engineers
**IoT**

Internet of Things
**ISM**

Industrial, Scientific, and Medical band
**kNN**

K-Nearest Neighbor
**LORAWAN**

Long Range Wide Area Network
**LSNS**

Lubben Social Network Scale
**LSTM**

Long Short-Term Memory
**LTE**

Long Term Evolution
**MAC**

Medium Access Control
**MEQ**

Morningness-eveningness questionnaire
**ML**

Machine Learning
**MTR**

Mobile Telepresence Robots
**NA**

Negative affect
**NLP**

Natural language processing
**NN**

Neural Network
**P2P**

Peer-to-Peer
**PA**

Positive affect
**PANAS**

Positive and Negative Affect Schedule
**PHQ9**

Patient Health Questionnaire
**PPG**

Photoplethysmography
**QoS**

Quality of Service
**RBF**

Radial Basis Function
**REBT**

Rational Emotive Behavior Therapy
**RF**

Radio Frequency
**RNN**

Recurrent Neural Network
**ROC**

Receiver operating characteristic curve
**SDK**

Software Development Kit
**SpO2**

Peripheral capillary oxygen saturation
**SVM**

Support-Vector Machine
**UCLA**

University of California, Los Angeles
**UWB**

Ultra Wide-Band
**VR**

Virtual Reality
**WAN**

Wide Area Network
**WHO**

World Health Organization
**WiFi**

Wireless Fidelity
**WLAN**

Wireless Local Area Network

## Figures and Tables

**Figure 1 sensors-22-01108-f001:**
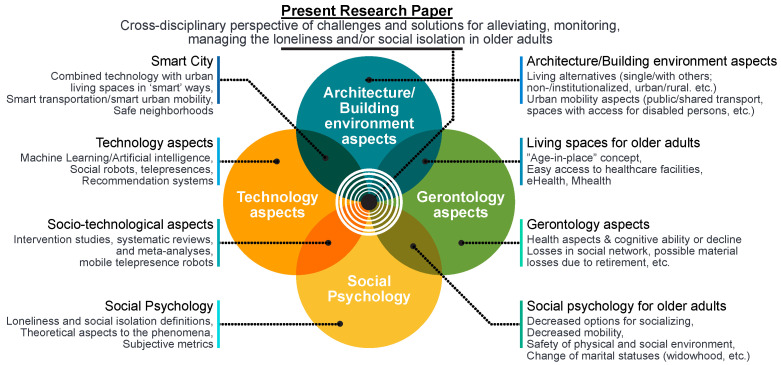
Venn diagram of the four dimensions of loneliness and social isolation addressed in this survey.

**Figure 2 sensors-22-01108-f002:**
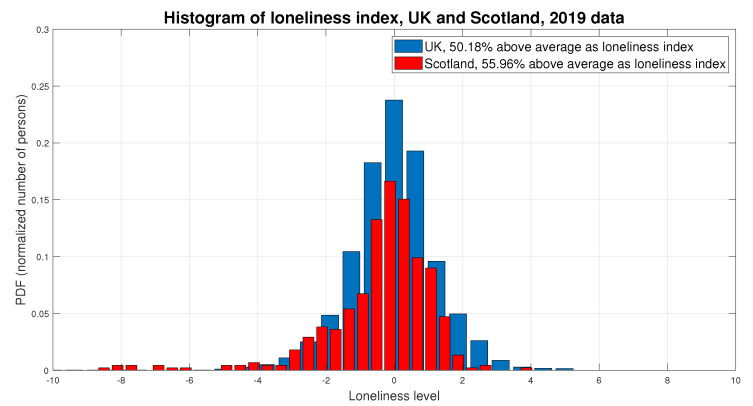
Examples of loneliness indices based on 2019 open-access data.

**Figure 3 sensors-22-01108-f003:**
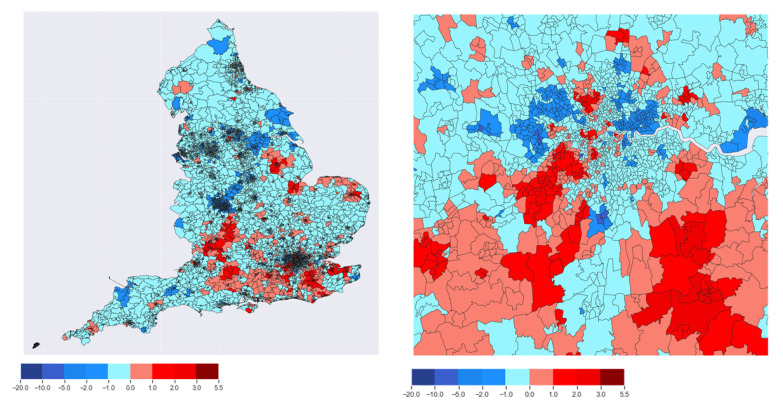
Heat maps indicating loneliness indices of England (**left**) and a zoomed in version of the London area (**right**) based on 2019 data . Here, lower indices indicate lower loneliness levels.

**Figure 4 sensors-22-01108-f004:**
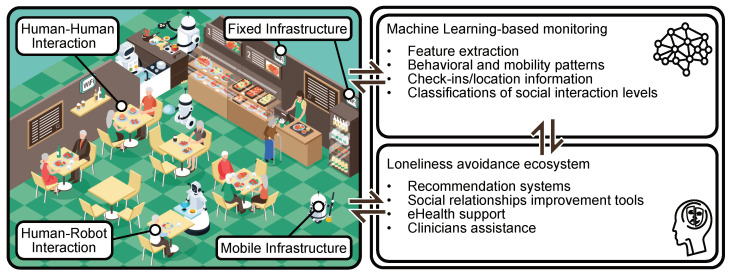
Top-level view of the proposed monitoring and management chain.

**Figure 5 sensors-22-01108-f005:**
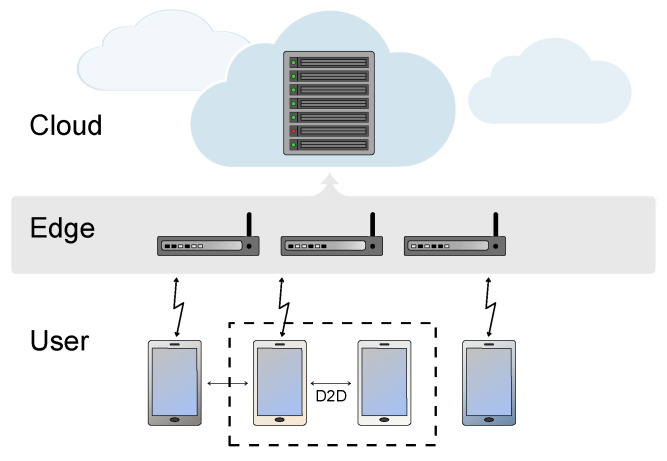
Example of the architecture involving proximity-based D2D communications in an Edge/Cloud environment.

**Figure 6 sensors-22-01108-f006:**
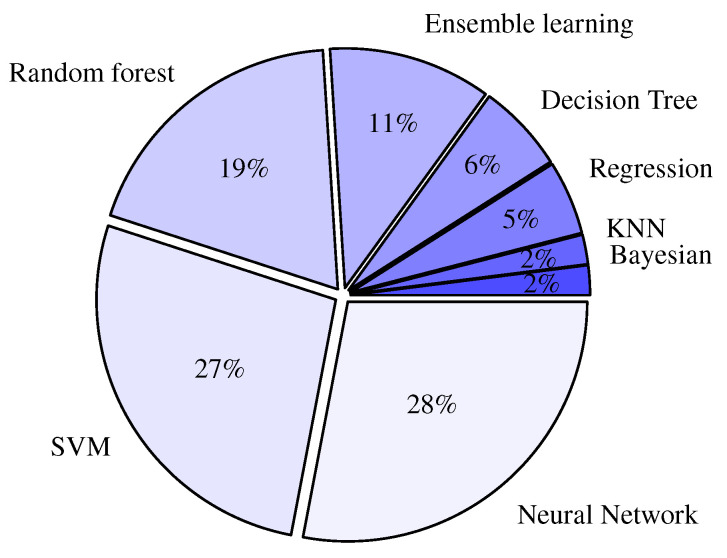
Distribution of the use of ML algorithms for eHealth applications, based on data from [140].

**Figure 7 sensors-22-01108-f007:**
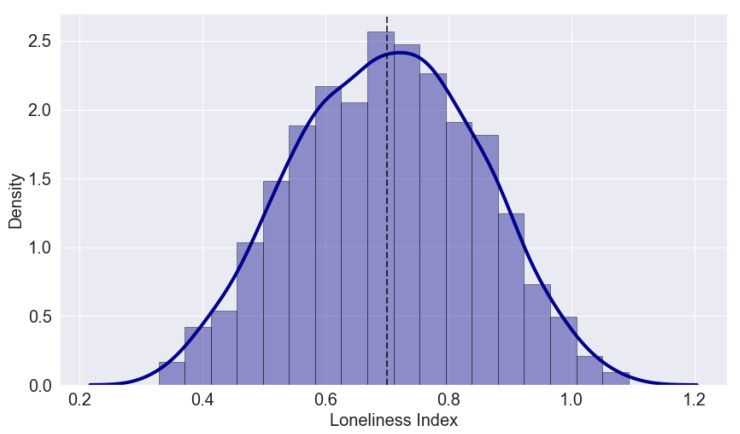
Distribution plot for loneliness index in the simulated scenario with dummy data.

**Table 1 sensors-22-01108-t001:** Related surveys in the literature and comparison with our survey.

References	Social Aspects in Loneliness	ML-Based Solutions for Loneliness	Built- Environment and Loneliness Aspects	Gerontology Aspects and Loneliness	Loneliness Metrics	Sensor Data for Loneliness Management
Hughes et al., 2004 [10]	●	❍	❍	❍	◗	❍
Hawkley and Cacioppo, 2010 [11]	●	❍	❍	❍	❍	❍
Ben-Zeev et al., 2015 [12]	❍	❍	❍	❍	❍	●
Chopik, 2016 [13]	◗	❍	❍	◗	◗	❍
Wilson, 2017 [14]	❍	❍	❍	◗	◗	❍
Badal et al., 2021 [15]	❍	●	❍	❍	❍	❍
Lam et al., 2021 [16]	●	❍	❍	❍	❍	●
Savage et al., 2021 [17]	●	❍	❍	●	❍	❍
Chau and Jame 2021 [18]	❍	❍	●	❍	❍	❍
Latikka et al., 2021 [19]	●	❍	❍	◗	❍	◗
Current survey	●	●	●	●	●	●

●—topic addressed in detail, ◗—topic partially addressed, ❍—topic not addressed

**Table 2 sensors-22-01108-t002:** Loneliness predictors.

Loneliness-Related Metrics	Description and Applicability
Anthropocentric data	Anthropocentric data include information about a person’s age, gender, height, and weight. According to [15], anthropocentric data could be useful in distinguishing several emotions, such as loneliness, sadness, and fear, in men and women.
MEQ value	The morningness-eveningness questionnaire [47] can be used to measure the person’s circadian rhythm to produce peak alertness in the morning or evening. MEQ as a predictor of social anxiety was studied in [90], and loneliness relationship to social anxiety was studied in [91].
Anxiety level	This parameter can be used to measure trait and state of anxiety. It can also be used to diagnose anxiety and distinguish it from depressive syndromes. Reference [91], for example, studied social anxiety as a significant predictor of loneliness.
Stress level	This parameter indicates potentially stressful events experienced by person. The causal and correlative links between stress and loneliness were studied, for example, in [92].
PANAS value	The PA and NA are dimensions to measure affective experience. PA and NA are found to be strongly related to extraversion and neuroticism personality factors, respectively [47]. Neuroticism and extraversion were shown to influence loneliness levels in [93].
Activity level	This can be used to indicate the person’s daily activity. Various activities could be sitting, walking, studying, eating, etc. Activities, such as sitting, lying down, sleeping, etc., which are often performed in a state of low energy consumption, are sedentary activities. A low level of objective physical activity and a high level of sedentary behavior was found to be correlated with higher social isolation and loneliness in older adults, in [61].
Heart-rate data	Heart rate data can be used to assess the inter-beat intervals variability in the time domain, frequency domain, and non-linear domain. It could be useful in giving an indirect index of the autonomous nervous system; hence, it is indirectly associated with the feeling of loneliness. The relationship between the heart-rate variability and chronic loneliness was studied, for example, in [94], but only for young women. Another study of the relationship between heart rates and loneliness was also performed in [95], also for young adults. Similar studies in older adults are still missing from the current literature.
Accelerometer data	Accelerometer data can be obtained from the positional sensors. The x,y,z components and the time and frequency domain components extracted from accelerometers can be useful in tracking a person’s activity. The activity levels, as discussed a few rows above, can be associated with loneliness levels, and lack of activity can be a predictor of loneliness [61].
Sleep quality index	This parameter measures the quality and patterns of sleep and is directly associated with perceived loneliness. This index measures sleep quality in seven subjective domains: sleep quality, sleep latency, sleep duration, habitual sleep efficiency, sleep disturbances, use of sleep medication, and daytime dysfunction [47]. The association between sleep quality and loneliness was studied, for example, in [96], for older adults in rural China. A higher quality of sleep was found to be positively correlated with a lower level of loneliness.
Proximity data	This data, based, for example, on the positioning or proximity-detection sensors able to estimate the distance between any two persons, can be used to estimate the social interactions among individuals. A study in [97] investigated BLE-based proximity detection, as well as other mobile data, as metrics for loneliness recognition; ML algorithms were used, and prediction accuracies around 71% were achieved.
Social-network diversity	The diversity of one’s social network (i.e., network size, level of engagement, etc.) was studied to be also correlated with feelings of loneliness and social isolation in [98]. Various social-network features are measurable through a variety of wearables and other IoT sensors, as described in Section 5.

**Table 3 sensors-22-01108-t003:** Examples of wearable devices for measuring loneliness-related metrics.

Brand & Type	Principle of Operation	Cost, €	Input Interface	Lifetime	Weight, Grams	Target Body Part	Type	Measured Parameter	Data Extraction Interface
Pozyx system	UWB, Accelerometer	≈100	Push button	20 months	21	Neck, wrist	Tag & AP	3D indoor spatio-temporal data	Pozyx Cloud service; CSV
Oura Ring	Temperature, Heart rate	≈300	–	3–4 days	6	Fingers	Ring	Heart rate, sleep pattern, temperature	Oura Cloud service; CSV
MiniFinder Pico	GPS	≈150	Push button	1 week	35	Pocket, neck	Tag	Outdoor spatio-temporal data	Remote API access through web services
Moodmetric Ring	EDA	≈500	–	4–7 days	6–10	Fingers	Ring	Stress levels, sleep pattern	Moodmetric app, Cloud services, and API
Withings ScanWatch	PPG, Accelerometer, ECG, EDA, Temperature, Pedometer	≈300	Rotating Crown	30 days	83	Wrist	Smart watch	Heart rate, activity, sleep, breathing disturbances	Moodmetric app, Cloud services, API
Imosi Smart Bracelet P11	PPG, Accelerometer, ECG, EDA, Temperature, Pedometer	≈70	GUI	6–7 days	26	Wrist	Smart watch	Heart rate, blood pressure, activity, sleep, breathing disturbances	P11 app, Cloud services, API
Fitbit Luxe	Heart rate, SpO2, sleep patterns, breathing rate, skin temperature	≈150	GUI	4–5 days	16	Wrist	Smart watch	Heart rate, activity, sleep pattern, stress	Fitbit app, Cloud services, API
Garmin Instinct	GPS, heart rate, blood oxygen, sleep, activity	≈200	GUI	1–3 days	52	Wrist	Smart watch	Heart rate, activity, outdoor spatio-temporal data, stress	Garmin Explore app, Cloud services, API

**Table 4 sensors-22-01108-t004:** Showcasing a few relevant technologies for D2D, Edge, and Cloud computing [124,125,126,127,128,129,130,131].

	Proximity-Based D2D			Edge Computing			Cloud Computing	
Network	Short-Range P2P			Short-Range WLAN			Long-Range	
Wireless technology	**BLE (v5.3)**	**WiFi Direct**	**LTE direct (3GPP Rel.12)**	**WiFi-4 (IEEE 802.11n)**	**WiFi-5 (IEEE 802.11ac)**	**WiFi-6 (IEEE 802.11ax)**	**LTE/4G (3GPP Rel.8)**	**5G (3GPP Rel.16)**
Frequency Band (GHz)	2.4	2.4; 5	0.45–3.7	2.4; 5	5	2.4; 5; 6	0.45–3.7	< 1; 1–7; 24–29
Channel Bandwidth (MHz)	2	20	1.4, 3, 5, 10, 15, 20	20, 40	20, 40, 80, 160	20, 40, 80, 160	1.4, 3, 5, 10, 15, 20	Up to 100
Channel Access Method	FH-CDMA, CSMA/CA, TDMA	CSMA/CA, SDMA	OFDMA, SC-FDMA	SDMA	CSMA/CA, SDMA	OFDMA	OFDMA, SC-FDMA	OFDMA
Expected data rate (Mbit/s)	2	250	100–300	600	6900	9600	300	10,000
Relative Latency	Average	Average-Low	Average-Low	Low	Ultra Low	Low	Low	Ultra Low
Reliability	Not guaranteed (ISM band)		Low (depends on the network awareness)	Not guaranteed (ISM band)			High (cellular operator-guaranteed)	

**Table 5 sensors-22-01108-t005:** Summary of relevant ML algorithms and their applicability for loneliness measuring and management.

ML Algorithm	Refs.	Applicability	Benefits	Challenges
Bayesian classifier	[141]	Building recommendation systems, combined with collaborative filtering approaches	Performs well with the categorical data	Requires a set of independent features which may be hard to acquire
Decision trees	[141]	Mental health and loneliness prediction	Generates easy-to-explain models and handles missing values well	With larger and complex datasets, it requires more time to converge and suffers from higher complexity
Ensemble learning	[142,143]	Recommendation systems and prediction of loneliness levels	Improves the generalization capacity of the model and makes predictions using data-fusion techniques for multiple data sources	Handling of accuracy and diversity among the individual models in an ensemble and handling high numbers of the members used for constructing an ensemble are difficult
Logistic Regression	[48]	Classification of older adults into loneliness classes, recommendation systems	It performs well with linearly separable and simple datasets	This algorithms not converge well for non-linear problems
NN	[141]	Loneliness prediction using time and frequency domain feature set from sensor data, recommendation systems	Good performance for complex datasets and non-linear problems	It requires a significant amount of training data and may lead to over-fitting and generalization
Random Forest	[15]	Classification loneliness levels using different sensor data, recommendation systems	Works well with categorical and numerical values	It is not easy to interpret for larger datasets
SVM	[15,141]	Loneliness monitoring based on selected features, recommendation systems	Widely used, typically good performance in classification with low number of classes (e.g., two class problem of high-level versus low-level of loneliness)	SVM may be not very suitable for very large and very noisy datasets and may under-perform in such cases

**Table 6 sensors-22-01108-t006:** Results of ML algorithms for conceptual model of loneliness prediction.

ML Algorithm	Accuracy	Precision	Sensitivity	ROC-AUC
Logistic Regression	99.2%	99.8%	98.6%	1
Random Forest	99.7%	99.8%	99.7%	1
SVM	97.3%	97.6%	96.7%	0.998

**Table 7 sensors-22-01108-t007:** Summary of the main identified challenges in loneliness management and monitoring in older adults.

Challenge	Groups	Refs.	Observed existing approach
Change of the computing paradigm/service in a seamless manner	T	[187]	Application of ML strategies with improved awareness
		[188,189]	Integrated software enablers for scheduling and technology selection
		[190,191]	Implementation of on-the-fly digital twin deployment approaches
Energy consumption-aware data processing	T	[192]	Lightweight technique for on-the-fly data encryption with pre-processing
		[193]	Energy-aware wearable sensing strategies
		[194]	Activity recognition-based strategy for adaptive compression
Lack of network and system resources	T	[195,196]	Offloading via proximity-based P2P network
		[197]	D2D strategies based on multi-cast for improving QoS
		[198]	Utilization of approximate computing techniques for computing resources identification
Challenges related to ML utilization; also see Table 5	T	[199]	ML-based authentication in IoT systems
		[200]	Ultra-low-power on-chip training and inference commands for power and computational efficiency for ML operation enablers
		[201]	Reduction of the overall execution time for classification problems, anomaly detection, etc.
Security and privacy-related aspects	T	[202]	Advance asymmetric encryption-based protocols
		[203,204]	The use of lightweight crypto-primitives to reduce the CPU load
		[205]	Integration of device/primitive-specific accelerators
		[206]	Identification new thresholds to fulfill application-specific security/privacy demands
Subjective interpretations may be hard to quantify or measure	S	[207]	Loneliness as a complex and multidimensional problem
		[207]	Understanding loneliness from a social-psychology perspective
		[208]	Combating loneliness with nostalgia
Dealing with cognitive impairments at older age	G	[209]	Creating supportive conditions which reduce the demand for controlled processing; training recollection, etc.
		[210]	aerobic exercise and dietary approaches
Ambiguous and heterogeneous intervention studies with elderly	G	[180]	Combined community-based and individual approach for interventions
		[178]	Theory-based approaches for social relationships and skill-building interventions
Avoiding ageism and age-segregation problems	A, T	[174]	Multiple benefits when inter-generational integration is achieved in urban/smart dwellings
		[211]	Implementation of ICT-based solutions for aging-in-place
Good access to healthcare while aging-in-place	A, G	[52,53]	Sense of place and sense of belonging, as well as known, lifetime friends and neighbors, can diminish feelings of loneliness; however, the same healthcare facilities as in an institutionalized environment may not be available
Choice of loneliness metrics among the existing ones	S, T	[10,78,79]	UCLA loneliness scale
		[80,81]	DJGLS loneliness scale
		[82,83]	LSNS loneliness scale
Trade-off between resources—affordability of a living space and social sustainability	A, S, T	[59]	Use of ICT and AI for achieving best trade-offs
		[212]	Integrating social-justice concept for increased well-being
Taking into account the time variability and other dynamic behavior	S, T, G	[213]	Chronic versus acute social isolation and loneliness
		[214]	Acute loneliness & Rational Emotive Behavior Therapy (REBT)
Attain a close-to-zero learning barrier	S, T, G	[215]	Life-time engagement in stimulating learning activities
		[216]	Access to better instructions and support in using ICT tools
Lack of underlying models relating sensor data to loneliness levels	T, S, A, G	[90,91]	MEQ, social anxiety, and loneliness inter-dependencies
		[208]	Nostalgia & loneliness
		[93]	Neuroticism, extraversion, and loneliness
		[61]	Physical activity and loneliness
		[94,95]	Heart rate and loneliness

A—Architecture; T—Technology; S—Social Psychology; G—Gerontology.

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
