# Peer review of "Managing Perceived Loneliness and Social-Isolation Levels for Older Adults: A Survey with Focus on Wearables-Based Solutions"

_sensors, 2022, doi:10.3390/s22031108_

Round 1

Reviewer 1 Report

This article presents several aspects concerning monitoring and reducing loneliness and is more appropriate to review the article. The authors should add more detail for the proposed solution and experimental data.
R13: The authors should replace the acronym 'a.k.a.' with 'also known as'.
R763: It is not very clear the architecture of the wearable device. What type of sensors are integrated?
R767-768: What type of data collected from sensors is used in ML applications?
From Table 3 which technologies are used for the proposed wearable device?
R887-R912:What type of ML algorithm was used for the proposed solution-based wearable device for loneliness monitoring?
R913: Maybe table 4 should be added for comparison of the benefits/challenges of the ML algorithm used for the proposed wearable system.
A comparison of the proposed solution (wearable system+ NL algorithm for classifying/monitoring loneliness) is missing.
The conclusion part and future perspective are missing.
R1192: The horticultural therapy and real animal/pet robots interaction have relevance in the proposed solution (wearable device-based ML algorithm) for monitoring loneliness?

Author Response

Please find our answers to the Reviewers' comments in the attachment.

Reviewer 2 Report

The paper contains a valuable contribution. The subject is within the scope of the journal and the objective of research is well stated. However, some clarifications about the underlying hypothesis / scope are needed.

In the opinion of this Reviewer the manuscript deserves to be published once the Author takes into account the raised issues.

Introduction / Literature review

  1. The research scope is clear as well as the literature review. Anyway, the authors should better highlight the innovative aspects of their work in the manuscript.

What are the advantages / findings in the proposed paper, which are not covered by other studies/reviews?

Wearable sensors for measuring loneliness and/or isolation levels

  1. Table 2: it is not clear to this reviewer the functions of the single apparatus exposed. My suggestion is to add two different columns: one for the sensors and the measurement the apparatus is capable of; the second one with the interface for data exchange.
  2. An important aspect to be cited in the “Data extraction and manipulation” section is the synchronization among the different boards/apparatus used for the vital and ambiental parameters.

Synchronization is an important issue in such a system and a lot of papers analyze it in detail.

How much does synchronization problem could affect the entire system? I assume that in order to correlate different measures coming from different acquisition system, the timestamp of the acquisition is crucial.

Could your system obtain better results with a synchronization algorithm such as https://doi.org/10.1109/ACCESS.2021.3115440?

Relationship between wireless technologies and architectural design practices

  1. Table 3: please check the technologies described. For example, the Bluetooth 4 is classified as BLE. Moreover, BLE is too generic. For example, there are different data rate and channel bandwidth in relation to the BLE version. The BLE 5.0 has maximum datarate of 2Mbps even if the real throughput is much lower. While I’m not sure that the datarate of the BLE 4.x is about 3Mbps.

Minor

  1. The authors should check that all the used acronyms are explained and not repeated every time (e.g. PPG).
  2. Mainly the English is good and there are only a few typos. However, the paper should be carefully rechecked.

Author Response

(The authors gave the same response as above.)

Round 2

Reviewer 1 Report

I recommend this paper for publication.

Reviewer 2 Report

The authors have enriched their work. Many points may not have been clear and have not been addressed in a proper way.